# SPP-RL: State Planning Policy Reinforcement Learning

## Abstract

We introduce an algorithm for reinforcement learning, in which the actor plans for the next state provided the current state. To communicate the actor output to the environment we incorporate an inverse dynamics control model and train it using supervised learning. We train the RL agent using off-policy state-of-the-art reinforcement learning algorithms: DDPG, TD3, and SAC. To guarantee that the target states are physically relevant, the overall learning procedure is formulated as a constrained optimization problem, solved via the classical Lagrangian optimization method. We benchmark the state planning RL approach using a varied set of continuous environments, including standard MuJoCo tasks, safety-gym level 0 environments, and AntPush. In SPP approach the optimal policy is being searched for in the space of state-state mappings, a considerably larger space than the traditional space of state-action mappings. We report that quite surprisingly SPP implementations attain superior performance to vanilla state-of-the-art off-policy RL algorithms in a wide class of robotic locomotion environments.

## 1 Introduction

Research on reinforcement learning (RL) has brought a tremendous number of successful applications in diverse fields of science and technology. Application areas of RL can be split into two classes: discrete and continuous. Here, we are interested in continuous simulation environments, mostly in the robotics domain. Despite the magnitude of successful applications of RL in this domain, two of the main issues: sample efficiency and interpretability of RL trained agents, persist. These problems have utmost practical importance, especially for mission-critical applications. The current methods often require a vast amount of experience for training. The decision-making process of trained agents is not interpretable, sometimes resulting in finding proxy solutions. Thus, it is vital to research new RL algorithms that may partially solve the mentioned problems.

Traditionally, RL is based on the principle of searching for the optimal policy within the space of state-action mappings. We propose a new algorithm based on the principle of training an actor (a policy) operating entirely in the state space (state-state mappings). We call such policies the state planning policies (SPP), whose actions determine desired trajectories in the state-space. The task of training SPP may initially seem infeasible due to a significantly larger dimension of states than of actions. Nonetheless, quite surprisingly, we show that the approach is feasible and often leads to significant improvements in average performance for a class of robotic locomotion tasks.

We call our approach *State Planning Policy Reinforcement Learning (SPP-RL)*. It is a generic approach for problems specified using continuous environments. The main building block of SPP-RL – the RL agent can be implemented using virtually any RL algorithm. We chose to develop our approach using the state-of-the-art off-policy DDPG (Lillicrap et al., 2016), TD3 (Fujimoto et al., 2018), and SAC (Haarnoja et al., 2018a) algorithms. Note that, in SPP-RL we need another trainable model to communicate the policy output to the environment; as such, we incorporate a learnable inverse dynamics control model (IDM), see Fig. 1. The overall algorithm optimizes the policy simultaneously with CM. To ensure that the policy target states satisfy physical and under-actuation constraints, we formulate a constrained optimization objective for policy training. Our work lies within the category of RL methods that already implemented state-state policies, including work on hierarchical RL like Nachum et al. (2018), and the D3G algorithm from Edwards et al. (2020).

**Summary of results.**   Although the SPP-RL algorithm searches for the optimal policy within a much larger space, our performance benchmarks revealed that SPP-RL implementations often outperform their vanilla RL counterparts. Experiments performed in Ant & Humanoid tasks from MuJoCo suite (Todorov et al., 2012; Brockman et al., 2016) reveal that SPP-DDPG outperforms DDPG. Experiments in Safety-Gym Level 0 environments (Ray et al., 2019) demonstrate that SPP-TD3 and SPP-SAC outperform by a great margin TD3 and SAC, respectively. Experiments in AntPush task (Nachum et al., 2018) show that SPP-TD3 outperforms hierarchical RL method HIRO (Nachum et al., 2018) and provides some interpretability of the agent behavior.

We hypothesize that the superior performance of SPP-RL in the tested continuous environments originates in more efficient state-space exploration by state-state policies than traditional state-action policies; here noise is being added to target states rather than actions. To argue this, we performed series of experiments, including evaluation of a shadow agent utilizing experience from SPP and vanilla replay buffers (Sec. 5.4) and a study of the distributions of states gathered in different replay buffers (App. E.2). To analyze which features of the algorithm are crucial for superior performance, we report on a thorough ablation study of SPP-TD3 (App. E.1).

We implemented SPP-RL methods as a modular PyTorch library shared as open-source. SPP-RL algorithms are derived from their vanilla RL counterparts, making extending the library with new RL algorithms straightforward. We also share videos with test episodes of the trained agents to accompany benchmark plots (web, 2021).

Last but not least, we demonstrate theoretical convergence of clipped Double Q-learning implemented in SPP-TD3 algorithm within the classical stochastic processes, finite-state/action spaces setting and in environments based on rigid body dynamics model, refer to App. B.3.

## 1.1   RELATED WORK

We present a (non-exhaustive) list of related works; refer to Tab. 1 for a perspective on related work. The closest approach to ours is the D3G algorithm introduced by Edwards et al. (2020), which includes state planning policies, and introduces a novel form of the value function defined on state-next state pairs. There are two main ways our method is distinct. First, SPP employs the classical formulation of the value function. Also, we do not include a forward dynamics model nor the cycle loss. Instead, to guarantee consistency of the policy target-states in SPP, we formulate a constrained optimization problem (compare Fig. 2) solved via Lagrangian optimization.

Our work builds on the classical RL algorithms going back to REINFORCE (Williams, 1992), Asynchronous Actor-Critic (Mnih et al., 2016), and especially the off-policy actor-critic algorithms including Q-Prop (Gu et al., 2017), DDPG (Lillicrap et al., 2016), SAC (Haarnoja et al., 2018a;b), and TD3 (Fujimoto et al., 2018). State planning policies have been used in hierarchical RL (HRL) methods like HIRO (Nachum et al., 2018) and FuN (Vezhnevets et al., 2017). Contrary to HRL SPP-RL approach does not employ a hierarchy of multiple policies nor state conditioned value functions.

Training predictive models (like IDMs) is fundamental for the model-based RL approach including algorithms: a locally linear latent dynamics model (Watter et al., 2015), model-based planning for discrete and continuous actions (Henaff et al., 2017), model-predictive control (Chua et al., 2018), and model based policy optimization (Janner et al., 2019). We deployed IDMs for mapping current-target states to actions; other applications of IDMs in RL include the context of planning: search on replay buffer (Eysenbach et al., 2019), episodic memory graph (Yang et al., 2020), and topological memory for navigation (Savinov et al., 2018). Existing many other applications of IDM in context of RL including: policy adaptation during deployment (Hansen et al., 2021), sim to real transfer (Christiano et al., 2016), adversarial exploration (Hong et al., 2020), and curiosity-driven exploration (Pathak et al., 2017).

## 1.2   BACKGROUND

Following the standard setting used in RL literature, we work with infinite horizon *Markov decision process* (MDP) formalism $(\mathcal{S}, \mathcal{A}, P, r, \rho_0, \gamma)$, where $\mathcal{S}$ is a *state space*, $\mathcal{A}$ is a *action space*, $P \colon \mathcal{S} \times \mathcal{A} \times \mathcal{S} \to [0, 1]$ is a *transition probability distribution*, $r \colon \mathcal{S} \times \mathcal{A} \to \mathbb{R}$ is a *reward function*, $\rho_0$ is an *initial state distribution*, and $\gamma \in (0, 1)$ is a *discount factor*. From now on we assume that the MDP is fixed. In RL the agent interacts with $E$ in discrete steps by selecting an action $a_t$ for the state $s_t$

Table 1: A Perspective on Related Work

| Technique | State-state policy | Inverse/forward model | State cond. $Q$ funct. | Policy Hierarchy | Planning horizon |
|---|---|---|---|---|---|
| SPP (ours) | yes | inverse | no | single policy | single step |
| D3G | yes | inverse | yes | single policy | single step |
| HRL | yes(upper level) | inverse | yes(upper level) | multiple policies | multiple steps |
| Planning | yes | inverse | yes | N/A | multiple steps |
| Model based | no | forward | N/A | single policy | single step |

at time $t$, causing the state transition $s_{t+1} = E(s_t, a_t)$, as a result the agent collects a scalar reward $r_{t+1}(s_t, a_t)$, the return is defined as the sum of discounted future reward $R_t = \sum_{i=t}^{T} \gamma^{(i-t)} r(s_i, a_i)$. The goal in RL is to learn a policy that maximizes the expected return from the start distribution.

## 2   STATE PLANNING POLICY REINFORCEMENT LEARNING APPROACH

Our SPP approach is rooted in state-state reinforcement learning, by which we mean setting in which RL agent is trained to plan goals in the state-space, the approach already employed e.g., in HRL, planning, D3G RL algorithms (see Tab. 1). In SPP a state planning policy $\pi$ given the current state $s_t$ outputs $z_t$ – the desired target state to be reached by the environment in the next step. Forcing the environment to reach the desired state requires translating the target state to a suitable action $a_t$. Hence, we employ an additional model capable of mapping the current state-target state pair $(s_t, z_t)$ to the action $a_t$ – a (trainable) IDM model. Ideally, we like to have consistency $z_t(s_t) \approx s_{t+1}$. The consistency cannot be guaranteed a-priori, is rather achieved in SPP setting by employing a constrained optimization approach. A diagram illustrating SPP approach is presented in Fig. 1.

We have freedom of choice of the particular RL algorithm (RL agent) and IDMs implementations. Currently, we use feed-forward neural networks, and RL Agent using implementations of the state-of-the-art off-policy RL algorithms: DDPG (Lillicrap et al., 2016), TD3 (Fujimoto et al., 2018) and SAC (Haarnoja et al., 2018a). We present details of SPP-RL implementation in Sec. 4 using as the example SPP-DDPG. The encountered experiences during the execution of an off-policy RL algorithm are stored in replay buffer $\mathcal{D}$.

The main building block of SPP-RL are the state planning policies, intuitively a state planning policy selects a desired trajectory in the state-space of the environment.

**Definition 1.** We call a *state planning policy* a map $\pi_\theta \colon \mathcal{S} \to \mathcal{P}(\mathcal{S})$ parametrized using a vector of parameters $\theta \in \mathbb{R}^{n(\theta)}$, and we denote $\pi_\theta(z|s)$ a probability of the desired *target state* $z \in \mathcal{S}$ for the given current state $s \in \mathcal{S}$.

We call a *deterministic state planning policy* a parametrized map $\pi_\theta \colon \mathcal{S} \to \mathcal{S}$, and we denote $\pi_\theta(s) = z$.

We assume that $\pi$ has continuous and bounded derivatives with respect to $\theta$. We will call state planning policy whenever it is clear from the context deterministic/stochastic and omit the parameter subscript $\pi = \pi_\theta$.

Besides the state planning policy (Def. 1) the second main building block of the overall SPP agent is a model for mapping the current state-target pair $(s_t, z_t)$ to suitable action $a_t$. Following the existing literature, we call such model the *inverse dynamics control model* (IDM), or simply the *control model*.

**Definition 2.** For a given MDP $(\mathcal{S}, \mathcal{A}, P, r, \rho_0, \gamma)$. Let $s, z \in \mathcal{S}$. We define the *control model*:

$$\mathsf{CM} \colon \mathcal{S} \times \mathcal{S} \to \mathcal{A}, \quad \mathsf{CM}(s, z) = a,$$

i.e. for the given two states CM computes the action $a$. We call $s, z$ the initial state and the target state respectively, where $a$ informally satisfies $\arg\max_{b \in \mathcal{S}} P(s, a, b) \sim z$ for stochastic $E$, or $z \approx E(s, a)$ for deterministic $E$.

Obviously, in order to work, SPP requires consistency of the target states generated by the policy with the actual next-states of the environment. We call this property the state consistency property (or simply consistency) of $\pi$, refer to Fig. 2. As it may be intractable to verify SPP for all possible

interactions in continuous environments, we are interested in guaranteeing the state consistency for the experiences stored in the replay buffer.

**Property 1.** Let $\mathcal{D}$ be a replay buffer, CM be an IDM and $\pi$ be a (SPP) policy. We say that $\pi$ has the *state consistency* property with threshold $d > 0$ if it holds that

$$\mathbb{E}_{\substack{(s_t, s_{t+1}) \in \mathcal{D} \\ z_t \sim \pi(s_t)}} \left[ \|s_{t+1} - z_t\|_2^2 \right] \leq d,$$

for deterministic $\pi$ we have $z_t = \pi(s_t)$. Given $(z_t, s_{t+1})$, we call distance $\|z_t - s_{t+1}\|_2^2$ the *state-consistency distance* (refer Fig. 2). We will often assume that $d$ is known from context and omit 'with threshold $d > 0$'.

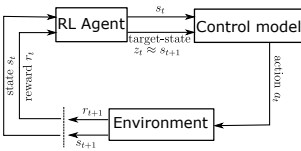

Figure 1: Diagram presenting our SPP method

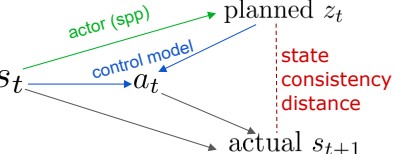

Figure 2: Diagram presenting the idea of state consistency property, ultimately we want to achieve $z_t \approx s_{t+1}$.

## 3    ENSURING STATE CONSISTENCY PROPERTY BY CONSTRAINED OPTIMIZATION

Our SPP algorithm is utilizing three parametrized models: IDM $CM_\psi$, policy model $\pi_\theta$, and $Q$-function model(s) $Q_\phi$. In our current implementation, all of the models are feed-forward neural networks. One way of ensuring the state consistency property (Prop. 1) is to modify the policy training loss, such that the expected values are maximized under fixed state consistency distance penalty. However, such an approach has many disadvantages, e.g., choosing appropriate learning temperature ($\lambda$) for the state consistency penalty is a very delicate issue, see the ablation study in Appendix E.1. It is easy to notice that setting its value too low would result in $\pi$ biased towards nonphysical target-states. On the other hand, setting its value too high would make $\pi$ overly conservative for off-policy states. Hence, we find a solution relying on constrained optimization more appealing for the studied problem. Namely, the objective for policy training is to maximize the sum of discounted rewards assuming a fixed threshold for the state consistency distance.

**Definition 3.** Let $\pi$ be a state planning policy (Def. 1), CM be a control model (Def. 2), $\mathcal{D}$ be the replay buffer with experience generated by executing an off-policy RL algorithm. In particular $z_t$'s are target states evaluated on-the-fly by $\pi$ (susceptible to be changed during the course of algorithm), and $s_{t+1}$'s are $E$ next-states. Let $d > 0$ be a fixed hyperparameter.

We define *the constrained objective for state planning policy* $\pi$ as follows

$$\max_{\pi} \; \mathbb{E}_{\substack{\tau \sim \pi, \text{CM} \\ a_i = \text{CM}(s_i, z_i)}} \left[ \sum_{i=0}^{T} \gamma^i r(s_i, a_i) \right], \tag{1a}$$

$$\text{s.t.} \; \mathbb{E}_{\substack{(s_t, s_{t+1}) \in \mathcal{D} \\ z_t \sim \pi(s_t)}} \left[ \|s_{t+1} - z_t\|_2^2 \right] \leq d, \tag{1b}$$

where $d$ is a hyperparameter for determining the allowed threshold for the expected divergence of predictions from actual next-states, equation 1a is an expectation over the policy trajectories generated by both of the state planning policy and IDM (trajectory is composed out of tuples $(s_t, z_t, a_t)$). The whole optimization process of equation 1a is being performed off-policy, see Sec. 4.

The constrained objective in equation 1 is being solved using the standard Lagrange multiplier method. The max-min Lagrangian objective $\mathcal{L}(\pi, \lambda)$ for the constrained optimization problem takes the form $\max_\pi \min_{\lambda \geq 0} \mathcal{L}(\pi, \lambda) = \mathbb{E}_{\tau \sim \pi}[R_0(\pi)] - \lambda \left( \mathbb{E}_{\mathcal{D}} \left[ \|s_{t+1} - z_t\|_2^2 \Big|_{z_t \sim \pi(s_t)} \right] - d \right)$. For more details refer to App. C.1.

## 4 ALGORITHM IMPLEMENTATION

We briefly present here details of the SPP Algorithm implementation, more detailed discussion can be found in Appendix C. We implemented SPP-DDPG, SPP-TD3, and SPP-SAC as a modular Python library within PyTorch framework (Paszke et al., 2019). All gradient optimization steps were performed using the Adam optimizer by Kingma & Ba (2014). Many of the algorithmic choices were motivated by the Spinning Up RL on-line resource (Achiam, 2018). We publish the modular SPP-RL software package as open-source (web, 2021). For illustrative purposes, we present the pseudo-code of the full SPP-DDPG algorithm in Algorithm 1. SPP-SAC and SPP-TD3 algorithms are presented in Appendix C. We provide theoretical derivations concerning SPP-RL algorithms, including convergence in the classical finite setting and policy gradient analogues to vanilla RL in Appendix B. We emphasize that SPP algorithms are not using any extra samples, i.e. the samples utilized for ICM training are added to the buffer and then reutilized for RL training, and if the buffer is full new samples are not being added anymore. An important caveat of our policy implementation in SPP-DDPG, not present in the vanilla DDPG, is that the output of $\pi$ is being normalized in order to reflect the physical bounds. Contrary to vanilla DDPG where $\pi$ outputs actions within well-defined uniform bounds, in SPP a suitable normalization of target state $\pi$ output is being computed online – depends on the past $E$ observations. Implementation of $\pi$ is a feed-forward neural network (refer to Appendix D for details) with tangential outputs bounded within $[-1, 1]$. Hence, we normalize the output of neural network $\pi$ by utilizing the current mean and min/max values of the past observations in replay buffer $\mathcal{D}$. We recompute mean and min/max values after each episode of the algorithm. Our base implementation of DDPG algorithm is parametrized by the usual hyper-

---

**Algorithm 1:** SPP-DDPG Algorithm

**input** : environment $E$; initial model parameters $\theta, \phi, \psi$; state planning distance threshold $d$; empty replay buffer $\mathcal{D}$; the DDPG algorithm hyperparameters

**output:** trained model parameters $\theta, \phi, \psi$; total return

**repeat**
 Sample random action $a \sim \mathcal{U}$;
 Store experience $(s_t, a_t, z_t = s_{t+1}, r_{t+1}, s_{t+1})$ in replay buffer $\mathcal{D}$; (use next-state as the initial actor actions)
**until** *random exploration is done*;
**repeat**
 **if** *buffer $\mathcal{D}$ is not full* **then**
  Compute actor prediction $z_t = \pi(s_t) + \varepsilon$, where $\varepsilon \sim \mathcal{N}$;
  Compute action $a_t = \mathsf{CM}(s_t, z_t)$ and observe reward $r_{t+1}$ and next state $s_{t+1}$;
  Store experience $(s_t, z_t, a_t, s_{t+1}, r_{t+1})$ in $\mathcal{D}$;
 **end**
 **if** *it's time to update* $\mathsf{CM}$ **then**
  Sample $\{b_i = \{((s_t, s_{t+1}), a)\}\}_{i=1}^{b}$ $b$ batches of samples from replay buffer $\mathcal{D}$;
  SGD train $\mathsf{CM}$ using the batches and MSE loss;
 **end**
 **if** *it's time to update actor and critic* **then**
  **for** *update steps* **do**
   Randomly sample $\mathcal{B} = \{(s_t, z_t, a_t, s_{t+1}, r_{t+1})\}$ set of batches from $\mathcal{D}$;
   Compute $\tilde{a}_{k+1} = \mathsf{CM}(s_{t+1}, \pi(s_{t+1}))$;
   Compute targets $y = r_{t+1} + \gamma Q^{\pi,\mathsf{CM}}_{\phi_{targ}}(s_{t+1}, \tilde{a}_{k+1})$ (using target parameters $\phi_{targ}$);
   Update $\phi = \phi - \frac{l_\phi}{|\mathcal{B}|} \cdot \nabla_\phi \sum_{\mathcal{B}} \left(y - Q^{\pi,\mathsf{CM}}_\phi(s_t, a_t)\right)^2$;
   Update policy parameters (ascent w.r.t $\theta$ of max-min Lagrangian obj.) $\theta = $
   $\theta + l_\theta \left( \frac{1}{|\mathcal{B}|} \cdot \nabla_\theta \sum_{\mathcal{B}} Q^{\pi,\mathsf{CM}}_\phi(s_t, a_t) \Big|_{a_t = \mathsf{CM}(s_t, \pi_\theta(s_t))} - \frac{\lambda}{|\mathcal{B}|} \cdot \nabla_\theta \sum_{\mathcal{B}} \|s_{t+1} - \pi_\theta(s_t)\|_2^2 \right)$;
   Update (descent w.r.t. $\lambda$ of max-min Lagrangian obj.)
   $\lambda = \lambda + l_\lambda \left( \frac{1}{|\mathcal{B}|} \sum_{\mathcal{B}} \|s_{t+1} - \pi_\theta(s_t)\|_2^2 - d \right)$;
   Update actor & critic $\phi_{targ} = (1 - \tau)\phi_{targ} + \tau\phi$; $\theta_{targ} = (1 - \tau)\theta_{targ} + \tau\theta$;
  **end**
 **end**
**until** *convergence*;

---

parameters including episode length, update batch size, Polyak averaging parameter ($\tau$), actor and critic learning rate $l$, maximal episode length, number of test episodes, $\gamma$. To ensure that we perform minimization equation 6 within the domain of positive $\lambda$ values, we optimize the parameter of the softplus function. All relevant hyper-parameters are provided in Appendix D.

We performed a thorough ablation study of the different features implemented in SPP-TD3 algorithm including the Lagrangian multipliers method (and different values of fixed $\lambda$), target state normalization, state-state critic ($Q(s, s')$); we report results in Appendix E.1.

## 5 EXPERIMENTAL EVALUATION

To show the feasibility of our method, we performed experiments on a set of benchmarks using continuous environments, most of them having large space dimensions. We performed all of the reported experiments using the default vector state input. We show that SPP-RL implementations compare favorably to their vanilla RL counterparts. As our SPP approach differs considerably from the vanilla off-policy RL, we performed a thorough hyper-parameter sweep from scratch. We provide the hyper-parameter values from the actual SPP implementations in Appendix D. For the sake of presentation we share videos with example test episodes rendered using the trained actors and high-resolution benchmark plots. All of our experiments are reproducible, we share the sources, training, evaluation, and trained models online (web, 2021). All of the reported experiments were run using CPU only, and a single experiment was always run on a single CPU core, i.e., we have not performed collecting experience in parallel. The experiments were performed on an example machine: AMD Ryzen Tr. 1920X, 64 Gb RAM, Ubuntu OS 18.04. Example average time of execution of $10^6$ steps stands at SPP-DDPG 5hrs $58'$, DDPG 3hrs, $7'$, (SPP-)SAC 19hrs $20'$, SAC 11hrs.

### 5.1 CLASSICAL MUJOCO

First, to measure the performance of SPP-RL we run benchmarks in the MuJoCo control tasks (Todorov et al., 2012) from OpenAI gym (Brockman et al., 2016): *Ant* and *Humanoid*. The experiments were performed using solely the vector state input. We present the results in Fig. 3. We compare the performance of SPP implementations with vanilla RL implementations in terms of average test episodes return. We emphasize that SPP algorithms do not increase the overall sample efficiency of the RL procedure. The samples utilized for ICM training are added to the buffer and then reutilized for RL training, and if the buffer is full new samples are not added anymore (refer to Algorithm 1). The algorithm does not take advantage of a pre-trained ICM before the actual RL training, rather optimizes $\pi$ and trains ICM simultaneously. However, as reported in Fig. 3, quite surprisingly, SPP based RL algorithms perform on par (TD3 & SAC case) or significantly better (DDPG case) than corresponding vanilla RL implementations in the MuJoCo environments. Analyzing the case of (SPP-)DDPG Figs.3a, 3d reveals that SPP significantly outperforms the vanilla implementation, as the vanilla implementation does not converge on those envs (compare with Achiam (2018)). Analyzing the case of (SPP)-TD3 reveals that for Ant env (Fig. 3b) SPP-TD3 performs on par with vanilla TD3 and outperforms by far D3G – the most related method. For Humanoid env (Fig. 3e) we see that SPP-TD3 outperformed vanilla TD3, the visible high variance of vanilla TD3 caused by a single run, which did not converge, D3G do not converge in this env. There is a visible divergence of the results we obtained for D3G in Ant from those reported by Edwards et al. (2020). We used MuJoCo v.1.5 library version that is known to be compatible with OpenAI gym; it is known that using v.2.0 results in zeroing out a part of observation space (in Ant & Humanoid) (gym, 2019), which may alter the results.

### 5.2 SAFETY-GYM (LOCOMOTION TASKS)

In the next set of benchmarks, we use environments from the safety-gym suite by Ray et al. (2019). Currently, we employed only Level 0 environments (which does not involve the cost function for violating the safety). We find Level 0 tasks from the safety-gym suite as the perfect ground to study the performance of SPP-RL in robotic locomotion environments, the goal being to steer agents (robots) to solve planar goal-reaching tasks. Moreover, we concentrate on difficulties arising from higher dimensionality of the state (and actions) space rather than maximizing returns under safety constraints. The experiments were performed using solely the vector state input. We leave investigating

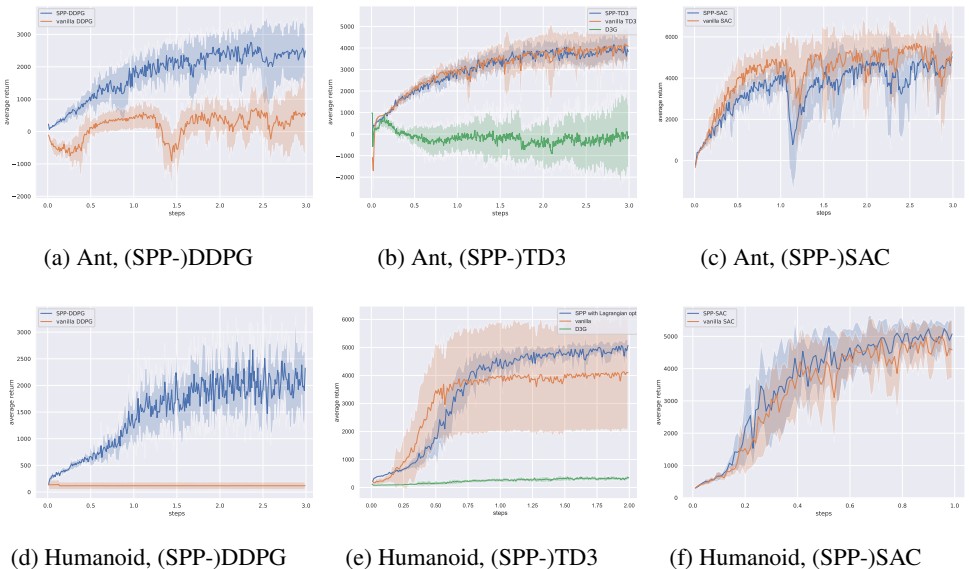

(a) Ant, (SPP-)DDPG          (b) Ant, (SPP-)TD3          (c) Ant, (SPP-)SAC

(d) Humanoid, (SPP-)DDPG     (e) Humanoid, (SPP-)TD3     (f) Humanoid, (SPP-)SAC

Figure 3: Experimental comparison of SPP implementations with corresponding vanilla off-policy RL on the two most complex MuJoCo benchmarks Ant (upper row) & Humanoid (bottom row). Figures show test return computed every 5k frames averaged over 10 different seeds. The continuous curve is the mean, whereas the faded color regions represent std. deviation. SPP RL algorithms in blue, corresponding vanilla RL in orange, only case of TD3 Fig. 3b, 3e include results obtained using D3G implementation in green. Refer to Appendix D for exact hyperparameters. Dimensions: Ant: $\dim(\mathcal{S}) = 111$, $\dim(\mathcal{A}) = 8$; Humanoid: $\dim(\mathcal{S}) = 376$, $\dim(\mathcal{A}) = 17$.

the higher-level environments considering the cost function as a topic of future research. We chose a subset of the most challenging Level 0 tasks, including Car-Push, Doggo-Goal, Doggo-Button environments. We also create a custom environment (termed Doggo-Columns) based on Doggo-Goal with additional 10 fixed pillars placed in the arena, obscuring the paths toward the goal.

We evaluate our SPP-TD3 implementation against state-of-the-art off-policy algorithms like TD3 and SAC. We do not include on-policy algorithms like PPO (Schulman et al., 2017) or TRPO (Schulman et al., 2015), as their performance in the studied environments is inferior to off-policy, refer to Ray et al. (2019) and the project web-page.

The results presented in Fig. 4 clearly show that SPP-TD3 is superior to vanilla off-policy algorithms within the studied safety-gym environments. Also, there is a noticeable difference in the learned behavior of the trained agents. The agents trained using the SPP-RL approach show smarter and more efficient behavior; for instance, the trained using SPP doggo robot learned an efficient gait of moving backward to mark a goal or press a button. For comparison, we publish videos of the trained agents online (web, 2021).

We argue that the performance boost exhibited by SPP-RL algorithms over vanilla counterparts is due to improved exploration in some sense. In Sec. 5.4 we show results from evaluating a TD3 shadow agent, i.e. vanilla TD3 agent utilizing for training some portion of experience from SPP-TD3 replay buffer. In Appendix E.2 we investigate differences in distribution of states collected by both of the methods.

## 5.3 HARDER EXPLORATORY TASK ANTPUSH

In order to demonstrate interpretability aspects of the SPP approach, we describe an experiment in AntPush environment from Nachum et al. (2018). The experiments were performed using solely the vector state input. The task is to control the ant such that it reaches the goal. The goal is hidden within a chamber behind a block. Therefore, Ant needs to learn to walk around the block and push it to the right first before eventually reaching the goal. Success is defined as finishing the episode within a radius 5 from the goal. We benchmark SPP-TD3 against the state-of-the-art

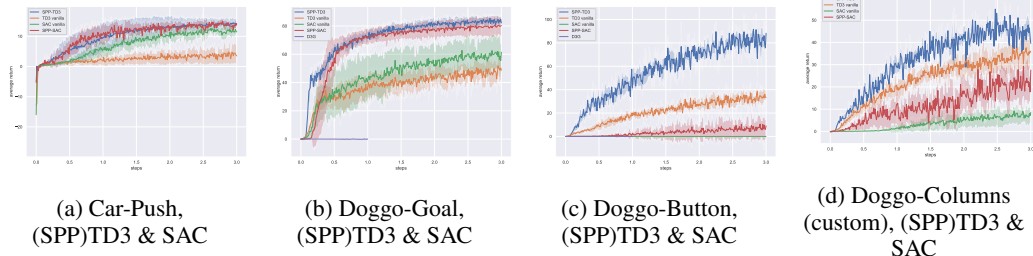

| (a) Car-Push, (SPP)TD3 & SAC | (b) Doggo-Goal, (SPP)TD3 & SAC | (c) Doggo-Button, (SPP)TD3 & SAC | (d) Doggo-Columns (custom), (SPP)TD3 & SAC |

Figure 4: Experimental comparison of SPP-TD3 with corresponding vanilla off-policy RL on set of safety-gym level 0 environments. Figures show test return computed every 5k frames averaged over 10 different seeds. The continuous curve is the mean, whereas the faded color regions std. deviation. SPP-TD3 algorithm blue, vanilla TD3 orange, SPP-SAC red, vanilla SAC green, D3G purple. D3G did not converge (return oscillated around zero, or it diverged in CarPush to a large negative score - removed from the plot for clarity). Refer to Appendix D for exact hyperparameters that we used to perform those experiments.

hierarchical RL HIRO method by Nachum et al. (2018). Specifically, we used the implementation by Qin (2021). Instead of reporting the achieved success rate of a single training run like in Nachum et al. (2018), which may be spurious if a lucky seed is chosen, we report the mean and std.dev. of the AntPush success rate using 10 random seeded training runs. Our experiments revealed that HIRO is highly susceptible to the random seed used. Only a single HIRO agent out of 10 trained using random seeds in total achieved a positive success rate, comparing to 7 out of 10 SPP-TD3 agents successfully learned to solve the task. The performance reported in Fig. 5a shows SPP-TD3 is eventually superior to HIRO. Example two solution paths are marked on Fig. 5b (blue curves). The right path is suboptimal as Ant blocks the entrance to the chamber where the goal is. The left path is optimal, Ant traverses to the left to push the red block away and open the passage towards the goal (green arrow).

Finally, Figs 5c, 5d show the obtained paths in the state space (blue dashed), and the policy target states $z_t$'s (orange solid), only the first two coordinates corresponding to the position of the Ant body in $x, y$ coordinates are illustrated. Observe that in the case of the suboptimal path in Fig. 5c, the planned path diverts to the left from the actual path (blue dashed), which indicates that the agent learned and attempted the correct behavior of pushing the red brick away and successfully open the entrance to the goal. In this case, however, the block is not movable; it is stuck as it was pushed forward before, hence as we see, the actual path in the state-space diverts in the middle. Figs 5c,5d show that apparently, the policy target states path being more erratic than the actual path in the state space. Erratic behavior can be mitigated by adjusting the hyperparameter $d$ in equation 1b (the smaller $d$, the closer the paths will be). Nonetheless, the policy target paths (orange) in Figs 5c,5d could be potentially used to cluster agent behavior, qualitatively differentiating two example agents executing (sub)optimal path. This information could then be used to pick appropriate agents for deployment.

## 5.4 More Efficient Exploration in SPP-RL

Our experimental evaluation using the safety-gym environments show that SPP-RL implementations outperform by a great margin their vanilla off-policy RL counterparts (TD3 and SAC) in terms of the average returns (See Fig.4). The performance boost is also visible in case of DDPG in MuJoCo Ant (Figs. 3a, 3d).

In this section, we argue the performance boost of SPP-RL compared to the vanilla RL counterparts. Our intuition is that exploration performed in the target-state space rather than in the action space may be more efficient in some cases. Exploration using SPP policies results in more viable experience being collected in the replay buffer, leading to more efficient Actor & Critic training. It is also possible that the constrained optimization induces some kind of curriculum. To confirm the mentioned intuition, we evaluated the performance of a TD3 shadow agent i.e., a vanilla TD3 Actor&Critic trained using (partially) experience collected by the SPP-TD3 agent. Both of the agents

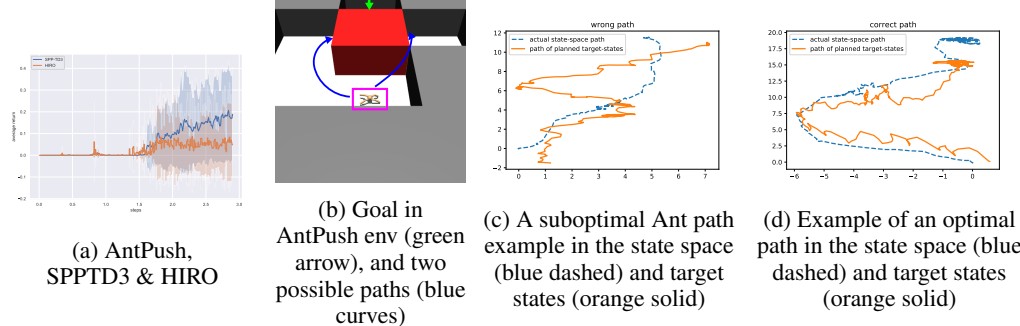

(a) AntPush, SPPTD3 & HIRO

(b) Goal in AntPush env (green arrow), and two possible paths (blue curves)

(c) A suboptimal Ant path example in the state space (blue dashed) and target states (orange solid)

(d) Example of an optimal path in the state space (blue dashed) and target states (orange solid)

Figure 5: Experiment in AntPush environment from Nachum et al. (2018). Fig. 5a shows success rate for SPP-TD3 & HIRO computed every 5k frames averaged over 10 different seeds (10 independent training runs were used). The continuous curve is the mean, whereas the faded color regions std. dev. Fig. 5b show two possible solution Ant paths. Figs 5c,5d show the paths in the state space (blue dashed) and target states (orange solid), the coordinates describing the position of the Ant body $(x, y)$ are used.

were trained in parallel. The TD3 shadow agent updates were performed using samples drawn from two of the replay buffers. The replay buffers of the SPP-TD3 and TD3 agent were used according to a 50/50 ratio. The results are presented in Fig. 6. Such TD3 shadow agent outperforms vanilla TD3, and eventually, its performance matches SPP-TD3 agent's in all of the studied safety-gym environments, excluding Doggo-Button.

Additionally, to address the question of how the distribution of experiences gathered by vanilla RL/SPP-RL compares to each other, we performed an additional experiment in Appendix E.2. We present plots of the discretized distributions of states encoded using a random encoder and cross-entropy of two distributions w.r.t. the quantity of gathered experience.

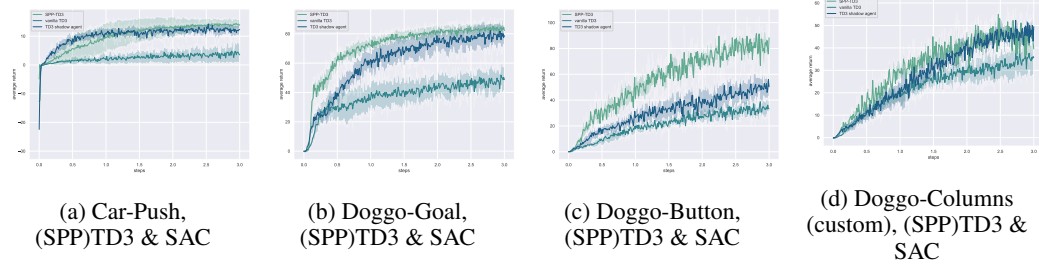

(a) Car-Push, (SPP)TD3 & SAC

(b) Doggo-Goal, (SPP)TD3 & SAC

(c) Doggo-Button, (SPP)TD3 & SAC

(d) Doggo-Columns (custom), (SPP)TD3 & SAC

Figure 6: Experimental evaluation of TD3 shadow agent, i.e. vanilla TD3 trained utilizing experience collected by a SPP-TD3 agent, on a set of safety-gym level 0 environments. Presented metrics are same as in Fig. 4. Shadow agent marked with the darkest color, vanilla TD3 and SPP-TD3 using lighter colors respectively.

## 6 CONCLUSIONS

We presented the State Planning Policy Reinforcement Learning (SPP-RL) – an approach for reinforcement learning, where the policy is selecting target states. Experiments performed on continuous benchmark environments often show the superior performance of SPP-RL compared to state-of-the-art off-policy RL algorithms. There are various avenues for future work pertaining to this research. One path is to include in our approach physically informed control models. Another, important path of work is to implement a long-term policy planning method scheme and application in the safety RL setting of SPP approach.

## 7 REPRODUCIBILITY STATEMENT

To ensure reproducibility of our results we published a supplementary material web page available at https://sites.google.com/view/spprl, there a link to a public github repository. The repository contains the sources, training & evaluation scripts, and trained models. The webpage also contains the benchmark plots in higher resolution, and also include videos with agents trained using different methods in the considered environments.

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

## A  APPENDIX

Videos visualizing the trained agents, and link to the git software repository with SPP-RL algorithms implementations are available online https://sites.google.com/view/spprl.

## B  THEORY OF STATE PLANNING POLICY RL ALGORITHMS

In this section we establish foundations of SPP-RL theory.

### B.1  STATE-VALUE AND ACTION-VALUE FUNCTIONS

In order to employ in SPP RL agents based on the state-of-the-art RL actor-critic algorithms, we start with defining the classical *state-value* and *action-value* functions in SPP setting.

Recall $R_t = \sum_{i=t}^{T} \gamma^{(i-t)} r(s_i, a_i)$ is the sum of discounted future rewards. The classical *Q-function* is

$$Q^\pi(s_t, a_t) = \mathbb{E}_{r_{i \geq t}, s_{i > t} \sim E, \, a_{i > t} \sim \pi}[R_t | s_t, a_t],$$

however, in our setting we use state planning policies (Def. 1) and we do not directly sample an action using the policy $a_i \sim \pi$. Instead we sample a target state $z_i \sim \pi$. Then the actual action $a_i$ is being computed $a_i = \mathsf{CM}(s_i, z_i)$.

Hence, the $Q$-function in SPP setting can take the form

$$Q^{\pi,\mathsf{CM}}(s_t, z_t) = \mathbb{E}_{r_{i \geq t}, s_{i > t} \sim E, \, z_{i > t} \sim \pi, \, a_i = \mathsf{CM}(s_i, z_i)}[R_t | s_t, z_t], \tag{2}$$

i.e. the expected value of following the trajectory planned by the policy $\pi$ in the state-space, and using IDM CM.

As we use *deterministic* IDM map $\mathsf{CM}\colon (s_t, z_t) \mapsto a_t$, we further reformulate equation 2 into

$$Q^{\pi,\mathsf{CM}}(s_t, a_t) = \mathbb{E}_{r_{i \geq t}, s_{i > t} \sim E, \, z_{i > t} \sim \pi, \, a_i = \mathsf{CM}(s_i, z_i)}[R_t | s_t, a_t]. \tag{3}$$

Analogously we define the *state-value* function $V^{\pi,\mathsf{CM}}$

$$V^{\pi,\mathsf{CM}}(s_t) = \underset{z_t \sim \pi}{\mathbb{E}} \left[ Q^{\pi,\mathsf{CM}}(s_t, \mathsf{CM}(s_t, z_t)) \right]. \tag{4}$$

Observe that if the parameters vector $\psi$ of CM is fixed, both of the formulations equation 2 and equation 3 are equivalent as CM maps $(s_t, z_t) \mapsto a_t$. However, in SPP CM is being trained, and its parameters are being changed during the execution of the full SPP algorithm. Therefore, we find the formulation equation 3 more suitable, as the established theory proves convergence of the *Bellman iterates* applied to equation 3 towards the optimal $Q$.

We evaluated the first variant of $Q$-function equation 2 in the ablation study presented in Appendix E.1 and it does not perform well. For the $Q$-function equation 3 we show theoretical convergence within the established framework in App. B.3.

### B.2  POLICY GRADIENT THEOREM

In this section, for the sake of completeness we include the fundamental *Policy Gradient Theorem* in the setting of SPP policies. We provide a formula for the gradient of the standard reward function in the deterministic policy gradient setting (Silver et al., 2014), which is applied in DDPG off-policy RL algorithm (Lillicrap et al., 2016), and also generalized in SAC off-policy algorithm (Haarnoja et al., 2018a) for stochastic policies.

Observe that the standard reward function in the deterministic policy gradient setting ($\pi$ and CM are deterministic) is given by

$$J(\theta) = \underset{s \sim d^\beta}{\mathbb{E}} \left[ Q^{\pi,\mathsf{CM}}(s, \mathsf{CM}(s, \pi(s))) \right], \tag{5}$$

where $d^\beta(s)$ is the stationary discounted state distribution.

Let the reward in the deterministic setting be given by equation 5. Assume CM is deterministic and differentiable, rest of assumptions like in (Silver et al., 2014, Thm. 1). Then, the formula for the standard policy gradient follows from (Silver et al., 2014, Thm. 1)

$$\nabla_\theta J(\theta) = \mathop{\mathbb{E}}_{s \sim d^\beta} \left[ \nabla_\theta \pi_\theta(s) \cdot \nabla_z \mathsf{CM}(s, z) \cdot \nabla_a Q(s, a)|_{z=\pi_\theta(s), \, a=\mathsf{CM}(s,z)} \right].$$

It is a straightforward extension of the formula given in (Silver et al., 2014, Thm. 1), applying one additional step of the chain rule.

Observe that in our algorithm we compute the policy gradient of the Lagrangian objective $\mathcal{L}(\pi, \lambda)$ in the deterministic setting, i.e.,

$$\mathcal{L}(\pi_\theta, \lambda) = J(\theta) - \lambda \left[ \mathop{\mathbb{E}}_{(s,s') \in \mathcal{D}} \left[ \|s' - \pi_\theta(s)\|_2^2 \right] - d \right],$$

where $\mathcal{D}$ is an experience replay buffer (given), and $J$ is the standard reward function in equation 5.

The gradient w.r.t. $\theta$ of the second term satisfies

$$\lambda \nabla_\theta \left[ \mathop{\mathbb{E}}_{(s,s') \in \mathcal{D}} \left[ \|s' - \pi_\theta(s)\|_2^2 \right] - d \right] = \lambda \mathop{\mathbb{E}}_{(s,s') \in \mathcal{D}} \left[ \nabla_\theta \pi_\theta(s) \cdot \nabla_z \|s' - z\|_2^2 \Big|_{z=\pi(s)} \right]$$

by applying the Leibniz integral rule to exchange order of derivative and integration, and continuity of $\pi_\theta$ and its derivative. In particular, note that the next state $s'$ is sampled from the buffer and thus does not depend on the policy.

Therefore, it holds that

$$\nabla_\theta \mathcal{L}(\pi_\theta, \lambda) = \mathop{\mathbb{E}}_{s \sim d^\beta} \left[ \nabla_\theta \pi_\theta(s) \cdot \nabla_z \mathsf{CM}(s, z) \cdot \nabla_a Q(s, a)|_{z=\pi_\theta(s), \, a=\mathsf{CM}(s,z)} \right] -$$
$$\lambda \mathop{\mathbb{E}}_{(s,s') \in \mathcal{D}} \left[ \nabla_\theta \pi_\theta(s) \cdot \nabla_z \|s' - z\|_2^2 \Big|_{z=\pi(s)} \right].$$

### B.3 CONVERGENCE OF Q-LEARNING IN SPP SETTING

In this section we argue convergence of Q-learning in SPP-TD3 algorithm within a finite setting using the existing framework of convergent stochastic processes based on the classic lemma by Bertsekas (2005). Used in Singh et al. (2000) to prove convergence of SARSA algorithm, and subsequently in Fujimoto et al. (2018) to prove convergence of TD3 algorithm Fujimoto et al. (2018). We provide the lemma here for completeness.

**Lemma B.1.** Consider a stochastic process $(\alpha_t, \Delta_t, F_t)$, $t \geq 0$, where $\alpha_t, \Delta_t, F_t \colon X \to \mathbb{R}$, which satisfies the equations

$$\Delta_{t+1}(x) = (1 - \alpha_t(x))\Delta_t(x) + \alpha_t(x)F_t(x), \quad x \in X, \, t = 0, 1, 2, \ldots$$

Let $P_t$ be a sequence of increasing $\sigma$-fields such that $\alpha_0$, $\Delta_0$ are $P_0$-measurable and $\alpha_t$, $\Delta_t$ and $F_{t-1}$ are $P_t$-measurable, $t = 1, 2, \ldots$. Assume that the following hold

1. the set of possible states $X$ is finite.

2. $0 \leq \alpha_t(x) \leq 1$, $\sum_t \alpha_t(x) = \infty$, $\sum_t \alpha_t^2(x) < \infty$ w.p. 1.

3. $\|E\{F_t(\cdot)|P_t\}\|_W \leq \kappa \|\Delta_t\|_W + c_t$, where $\kappa \in [0, 1)$ and $c_t$ converges to zero w.p. 1.

4. $\mathrm{Var}\{F_t(x)|P_t\} \leq K(1 + \|\Delta_t\|_W)^2$, where $K$ is some constant.

Then, $\Delta_t$ converges to zero with probability one.

The proof can be found e.g. in Jaakkola et al. (1994); Singh et al. (2000).

Based on Lemma B.1 Fujimoto et al. (2018) show the convergence of Clipped Double Q-learning to the optimal value function $Q^*$, as defined by the Bellman optimality equation w.p. 1.

**Theorem B.2** (Thm. 1 in Fujimoto et al. (2018)). Given the following conditions:

1. Each state action pair is sampled an infinite number of times.

2. The MDP is finite.

3. $\gamma \in [0, 1)$.

4. $Q$ values are stored in a lookup table.

5. Both $Q^A$ and $Q^B$ receive an infinite number of updates.

6. The learning rates satisfy $\alpha_t(s, a) \in [0, 1]$, $\sum_t \alpha_t(s, a) = \infty$, $\sum_t (\alpha_t(s, a))^2 < \infty$ with probability 1 and $\alpha_t(s, a) = 0$, $\forall (s, a) \neq (s_t, a_t)$.

7. $\mathrm{Var}[r(s, a)] < \infty$, $\forall s, a$.

Then Clipped Double Q-learning will converge to the optimal value function $Q^*$, as defined by the Bellman optimality equation with probability 1.

We can apply Thm. B.2 to show convergence of $Q$ learning in SPP-TD3 algorithm. First, observe that SPP-TD3 utilizes the same $Q$-functions, hence, the assumptions $2, \ldots, 7$ can be transferred to SPP-TD3 setting. However, the policy part is different.

Let us focus our attention to the assumption 1 of Thm. B.2. Observe that in the classical setting the assumption 1 is easily realizable by employing a policy performing either fully random exploration or $\varepsilon$-greedy exploration. In the studied SPP setting the situation is much more subtle. We do not have a-priori guarantees that the such an assumption is satisfied. Observe that even a randomly exploring policy may not realize such assumption. It will not be satisfied if for example the IDM is degenerate and maps each pair of states to the same action.

However, if we show

- for each pair of reachable states there is a unique action that results in the transition of the previous state to the next state,
- the SPP policy outputs only realizable target states.

then assumption 1 of Thm. B.2 can also be satisfied in SPP-RL setting by employing a fully random or $\varepsilon$-greedy SPP policy. The first condition above may not be true for general environments, as we can easily imagine an environment, where a particular state may be reached from another state by applying different actions. However, in the case of rigid body environments we can demonstrate that the desired property is true. We formalize this in the following lemma.

**Lemma B.3.** Let $E$ be a (deterministic) environment based on the rigid body dynamics model, i.e. the state transitions are computed using a time-wise discretization of a continuous and differentiable dynamics model. Let $s, s' \in \mathcal{S}$ be a pair of reachable states in environment $E$.

If there exists $a, a' \in \mathcal{A}$, such that $E(s, a) = E(s, a') = s'$, i.e. actions $a$ and $a'$ result in the same state transition, then it holds that $a = a'$.

*Proof.* It is known that the dynamics of rigid bodies is governed by the laws of kinematics and by the application of Newton's second law (Tsai, 1999). Namely, the equations governing the dynamics of rigid bodies in (time-dependent) generalized position-velocity coordinates $(q, v)$, $q \colon \mathbb{R} \to \mathbb{R}^n$, $v \colon \mathbb{R} \to \mathbb{R}^n$ take the following form

$$M(q)\frac{dv}{dt} = k(q, v) + \rho(q, v) + \tau,$$

where $M \colon \mathbb{R}^n \to \mathbb{R}^{n \times n}$ is the positive definite, symmetric inertia matrix, $k \colon \mathbb{R}^n \times \mathbb{R}^n \to \mathbb{R}^n$ is the vector of external and contact forces, $\rho \colon \colon \mathbb{R}^n \times \mathbb{R}^n \to \mathbb{R}^n$ is the sum of friction components, $\tau \colon \mathbb{R} \to \mathbb{R}^n$ is the applied external force (Pfeiffer & Glocker; Todorov et al., 2012).

Hence, there is a one-to-one mapping of state tuple $(q, v, \frac{dv}{dt}, \rho)$ and the external force $\tau$ (torque). The one-to-one mapping still holds for realizable environment $E$ based on simulation, where infinitesimal acceleration $dv$ is approximated by the finite difference $v_{t+1} - v_t$, $dt$ is a fixed time-step,

$\rho$ is calculated from the Cartesian body position and the fixed domain constraints, external force $\tau$ is calculated from input action $a$ by a linear transformation. □

We conclude with the following corollary demonstrating the convergence of $Q$ learning in SPP-TD3 algorithm within the finite setting.

**Corollary B.3.1.** Let

- $E$ be a (deterministic) environment based on the rigid body dynamics model, i.e. the state transitions are being generated by a time-wise discretization of a continuous and differentiable dynamics model,

- the experiences utilized for $Q$ updates be gathered using a random/$\varepsilon$-greedy SPP (deterministic) policy $\pi$,

- the target states form $\pi$ be realizable, i.e. if $z = \pi(s)$ for $s \in \mathcal{S}$, then it holds that $z \in \mathcal{S}$,

- IDM CM be implemented as a lookup table and receive infinite updates.

Moreover, assume $2, \ldots, 7$ of Thm. B.2.

Then, clipped Double Q-learning in SPP-TD3 will converge to the optimal value function $Q^*$, as defined by the Bellman optimality equation, with probability 1.

*Proof.* From Lemma B.3 it follows that for any pair of reachable states $s, s'$ of environment $E$ there exists a unique action $a$ resulting in transition $E(s, a) = s'$. Hence, as CM receives infinite updates and is implemented as a lookup table it will learn to map any of the policy target states (realizable states) to the unique action. It follows that each state action will be sampled infinite number of times in the limit. As the rest of Thm. B.2 assumptions are assumed we obtain the claim. □

## C  SPP-RL ALGORITHM DETAILS

### C.1  LAGRANGIAN OPTIMIZATION

To solve the constrained optimization problem we use the idea of Lagrange multipliers going back to the classical calculus. Lagrange multipliers were already applied in the RL context in Ray et al. (2019), where authors developed on-policy RL algorithms for safety RL. This method uses an adaptive penalty coefficient to enforce constraints. We find this method plausible, as it translates a constrained objective into solving a classical max-min problem that can be incorporated within any RL algorithm. The max-min Lagrangian objective $\mathcal{L}(\pi, \lambda)$ for the constrained optimization problem equation 1 takes the form

$$\max_{\pi} \min_{\lambda \geq 0} \mathcal{L}(\pi, \lambda) = \mathbb{E}_{\tau \sim \pi} [R_0(\pi)] - \lambda \left( \mathbb{E}_{\mathcal{D}} \left[ \|s_{t+1} - z_t\|_2^2 \Big|_{z_t \sim \pi(s_t)} \right] - d \right). \tag{6}$$

Intuitively, when the constraint equation 1b is satisfied, then the min w.r.t. $\lambda$ is attained for $\lambda = 0$, and causes $\lambda$ to decrease. On the other hand, if the constraint is not satisfied, then the min is attained for $\lambda = \infty$, and hence causing $\lambda$ to increase, the constraint satisfaction receives more weight in the overall max-min objective, eventually forcing the constraint to be fulfilled.

We solve the max-min Lagrangian objective for SPP policy using simultaneous gradient ascent w.r.t. the policy parameters $\theta$, descent w.r.t. the Lagrange multiplier $\lambda$, updates. Such an approach can be viewed as an instance of a primal-dual algorithm for constrained MDP. It is known in the literature that time-steps of Actor, Critic & dual variable gradient updates should be adjusted separately to guarantee convergence, refer to Borkar (2005) and references therein. The SPP algorithms use three learning rates: actor ($l_\theta$), critic ($l_\phi$), and dual variable ($l_\lambda$), which were adjusted using a hyper-parameter optimization. The actor and critic learning rates were equal in all of the studied algorithms, whereas the dual variable was fixed in all algorithms to $l_\lambda = 0.0001$.

## C.2  SPP-SAC Algorithm

The original DDPG algorithm, which is the base of SPP-DDPG presented in Algorithm 1 aims at training a deterministic policy, as contrary to for example SAC algorithm by Haarnoja et al. (2018a;b) that trains a stochastic policy. Its main principle is rooted in the very successful deep $Q$-learning method, originally applied for solving the Atari benchmark problem (Mnih et al., 2013). However, DDPG is suitable for continuous action spaces like in the MuJoCo tasks. SAC is an algorithm that combines ideas of DDPG and soft Q-learning Haarnoja et al. (2017) utilizing the advantages of both approaches, SPP-SAC is presented in Algorithm 2.

---

**Algorithm 2:** SPP-SAC Algorithm

---

**repeat**

    Sample random action $a \sim \mathcal{U}$;

    Store experience $(s_t, z_t, a_t, s_{t+1}, r_t)$ in $\mathcal{D}$; (use next-state as the initial actor actions)

**until** *random exploration is done*;

**repeat**

    **if** *buffer $\mathcal{D}$ is not full* **then**

        Sample actor prediction $z_t \sim \pi(s_t)$;

        Compute action $a_t = \mathsf{CM}(s_t, z_t)$;

        Observe reward $r_t$ and next state $s_{t+1}$;

        Store experience $(s_t, z_t, a_t, s_{t+1}, r_t)$ in $\mathcal{D}$;

        If $s_{t+1}$ is terminal, reset environment state;

    **end**

    **if** *it's time to update* $\mathsf{CM}$ **then**

        Randomly sample $\{b_i = \{(s_t, s_{t+1}), a_t\}\}_{i=1}^n$ $n$ batches of samples from replay buffer $\mathcal{D}$;

        SGD train $\mathsf{CM}$ using the batches and MSE loss;

    **end**

    **if** *it's time to update actor and critic* **then**

        **for** *update steps* **do**

            Randomly sample $\mathcal{B} = \{(s_t, z_t, a_t, s_{t+1}, r_t)\}$ set of batches from $\mathcal{D}$;

            Compute actions $\hat{a}_{t+1} = \mathsf{CM}(s_{t+1}, \hat{z}_{t+1})$, using samples $\hat{z}_{t+1} \sim \pi(z_{t+1}|s_{t+1})$;

            Compute targets $y = r_t + \gamma \min\limits_{i=1,2} Q_{\phi_{targ,i}}^{\pi,\mathsf{CM}}(s_{t+1}, \hat{a}_{t+1})$ ;

            Compute $\phi = \phi - l_\phi \cdot \nabla_\phi \frac{1}{|\mathcal{B}|} \sum_{\mathcal{B}} \left( y - Q_\phi^{\pi,\mathsf{CM}}(s_t, a_t) \right)^2$;

            Compute $\theta = \theta + l_\theta \cdot \nabla_\theta \frac{1}{|\mathcal{B}|} \sum_{\mathcal{B}} \big( \min\limits_{i=1,2} Q_\phi^{\pi,\mathsf{CM}}(s_t, \tilde{a}_{\theta,t}(s_t)) - \alpha \log \pi_\theta(\tilde{z}_{\theta,t}(s_t)|s_t)$,

             where $\tilde{z}_{\theta,t}(s_t) \sim \pi(z_t|s_t)$ is differentiable wrt $\theta$;

            Compute $\theta = \theta - \frac{l_\theta \lambda}{|\mathcal{B}|} \cdot \nabla_\theta \sum_{\mathcal{B}} \|s_{t+1} - \tilde{z}_{\theta,t}(s_t)\|_2^2$;

            Update $\lambda = \lambda + l_\lambda \left( \frac{1}{|\mathcal{B}|} \sum_{\mathcal{B}} \|s_{t+1} - \tilde{z}_{\theta,t}(s_t)\|_2^2 - d \right)$;

            Update target networks $\phi_{targ,i} = (1-\tau)\phi_{targ,i} + \tau\phi_i$, for i = 1,2;

        **end**

    **end**

**until** *convergence*;

---

## C.3  SPP-TD3 Algorithm

Twin Delayed Policy Gradient (TD3) (Fujimoto et al., 2018) is an improved DDPG algorithm using several tricks, which can be summarized as Clipped Double-Q Learning, Delayed Policy Updates, Target Policy Smoothing. SPP-TD3 algorithm is presented in Algorithm 3.

---

**Algorithm 3:** SPP-TD3 Algorithm

---

**input** : environment $E$; initial model parameters $\theta, \phi_1, \phi_2, \psi$; state planning distance threshold $d$; empty
       replay buffer $\mathcal{D}$; TD3 algorithm hyperparameters
**output:** trained model parameters $\theta, \phi_1, \phi_2, \psi$; total return
**repeat**
    Sample random action $a \sim \mathcal{U}$;
    Store experience $(s_t, a_t, z_{t+1}, r_{t+1}, s_{t+1})$ in replay buffer $\mathcal{D}$; (use next-state as the initial actor actions)
**until** *random exploration is done*;
**repeat**
    **if** *buffer $\mathcal{D}$ is not full* **then**
        Compute actor prediction $z_t = \text{clip}\left(\pi(s_t) + \varepsilon, z_{low}, z_{high}\right)$, where $\varepsilon \sim \mathcal{N}$;
        Compute action $a_t = \mathsf{CM}(s_t, z_t)$ and observe reward $r_{t+1}$ and next state $s_{t+1}$;
        Store experience $(s_t, z_t, a_t, s_{t+1}, r_{t+1})$ in $\mathcal{D}$;
    **end**
    **if** *it's time to update* $\mathsf{CM}$ **then**
        Sample $\{b_i = \{((s_t, s_{t+1}), a)\}\}_{i=1}^b$ $b$ batches of samples from replay buffer $\mathcal{D}$;
        SGD train $\mathsf{CM}$ using the batches and MSE loss;
    **end**
    **if** *it's time to update actor and critic* **then**
        **for** *update steps* **do**
            Randomly sample $\mathcal{B} = \{(s_t, z_t, a_t, s_{t+1}, r_{t+1})\}$ set of batches from $\mathcal{D}$;
            Compute target states $\widetilde{z}_{t+1}(s_{t+1}) = \text{clip}\left(\pi_{\theta_{targ}}(s_{t+1}) + \text{clip}(\varepsilon, -c, c), z_{low}, z_{high}\right)$, where
            $\varepsilon \sim \mathcal{N}(0, \sigma)$;
            Compute target actions $\widetilde{a}_{t+1}(s_{t+1}) = \mathsf{CM}(s_{t+1}, \widetilde{z}_{t+1}(s_{t+1}))$;
            Compute targets $y(r, s_{t+1}) = r + \gamma \min_{i=1,2} Q_{\phi_{targ,i}}^{\pi,\mathsf{CM}}(s_{t+1}, \widetilde{a}_{t+1}(s_{t+1})$ (using target
            parameters $\phi_{targ,1}, \phi_{targ,2}$);
            Update $\phi_1 = \phi_1 - \frac{l_\phi}{|\mathcal{B}|} \cdot \nabla_{\phi_1} \sum_{\mathcal{B}} \left(y(r, s_{t+1}) - Q_{\phi_1}^{\pi,\mathsf{CM}}(s_t, a_t)\right)^2$;
            Update $\phi_2 = \phi_2 - \frac{l_\phi}{|\mathcal{B}|} \cdot \nabla_{\phi_2} \sum_{\mathcal{B}} \left(y(r, s_{t+1}) - Q_{\phi_2}^{\pi,\mathsf{CM}}(s_t, a_t)\right)^2$;
            **if** *current_step mod policy_delay = 0* **then**
                Update $\theta = \theta +$
                $l_\theta \left( \frac{1}{|\mathcal{B}|} \cdot \nabla_\theta \sum_{\mathcal{B}} Q_\phi^{\pi,\mathsf{CM}}(s_t, a_t)\Big|_{a_t=\mathsf{CM}(s_t, \pi_\theta(s_t))} - \frac{\lambda}{|\mathcal{B}|} \cdot \nabla_\theta \sum_{\mathcal{B}} \|s_{t+1} - \pi_\theta(s_t)\|_2^2 \right)$;
                Update $\lambda = \lambda + l_\lambda \left( \frac{1}{|\mathcal{B}|} \sum_{\mathcal{B}} \|s_{t+1} - \pi_\theta(s_t)\|_2^2 - d \right)$;
                Update actor & critics $\phi_{targ,i} = (1-\tau)\phi_{targ,i} + \tau\phi_i$; $\theta_{targ} = (1-\tau)\theta_{targ} + \tau\theta$;
            **end**
        **end**
    **end**
**until** *convergence*;

---

# D HYPERPARAMETERS USED IN EXPERIMENTS

We present hyper-parameters used in our experiments in Table 2 for the SPP-DDPG, in Table 6 for the SPP-SAC. SPP-TD3 hyper-parameters were dependent on the set of benchmarks, and hyper-parameters specific for MuJoCo environments are in Tab. 3, for SafetyGym in Tab. 4, and for Ant-Push in Tab. 5.

In SPP-DDPG implementation we used the following architectures:

- Actor: $\dim(\mathcal{S}) \longrightarrow 256$ ReLU $\longrightarrow 256$ ReLU $\longrightarrow \dim(\mathcal{A})$ tanh;
- Critic: $\dim(\mathcal{S}) + \dim(\mathcal{A}) \longrightarrow 256$ ReLU $\longrightarrow 256$ ReLU $\rightarrow 1$;
- CM: $2 \cdot \dim(\mathcal{S}) \longrightarrow 100$ tanh $\longrightarrow 50$ tanh $\longrightarrow \dim(\mathcal{A})$ tanh;

In SPP-TD3 implementation we used the following architectures:

- Actor: $\dim(\mathcal{S}) \longrightarrow 256$ ReLU $\longrightarrow 256$ ReLU $\longrightarrow \dim(\mathcal{A})$ tanh;
- Critic: $\dim(\mathcal{S}) + \dim(\mathcal{A}) \longrightarrow 256$ ReLU $\longrightarrow 256$ ReLU $\rightarrow 1$;
- CM: $2 \cdot \dim(\mathcal{S}) \longrightarrow 100$ tanh $\longrightarrow 50$ tanh $\longrightarrow \dim(\mathcal{A})$ tanh;

| Hyperparameter | Value |
|---|---|
| $\gamma$ | 0.99 |
| $\tau$ | 0.005 |
| actor/critic learning rate $l_\phi = l_\theta$ | 0.0005 |
| $\lambda$ learning rate $l_\lambda$ | 0.0001 |
| episode length | 5000 |
| batch size | 100 |
| test episodes | 10 |
| update freq. & grad.steps. | 50 |
| exploration noise param. | 0.05 |
| replay buffer size | same as number of env. interacts |
| parameters specific for SPP-DDPG | |
| $d$ state consistency | 0.2 |
| CM hyper-parameters | |
| init. rand. samples | 20000 |
| learning rate | 0.005 |
| batch size | 128 |
| update frequency | 500 steps |
| update epochs | 1 |
| update batches | 100 |

Table 2: SPP-DDPG Algorithm hyperparameters, DDPG used the same hyperparameters, except exploration noise (set to 0.1).

| Hyperparameter | Value |
|---|---|
| $\gamma$ | 0.99 |
| $\tau$ | 0.005 |
| actor/critic learning rate $l_\phi = l_\theta$ | 0.0001 |
| $\lambda$ learning rate $l_\lambda$ | 0.0001 |
| episode length | 5000 |
| batch size | 100 |
| test episodes | 10 |
| update freq. & grad.steps. | 50 |
| exploration noise param. | 0.2 |
| policy noise | 0.2 |
| noise clip. | 0.5 |
| replay buffer size | same as number of env. interacts |
| parameters specific for SPP-TD3 | |
| $d$ state consistency | 0.2 |
| CM hyper-parameters | |
| init. rand. samples | 25000 |
| learning rate | 0.001 |
| batch size | 128 |
| update frequency | 500 steps |
| update epochs | 1 |
| update batches | 200 |

Table 3: SPP-TD3 Algorithm hyperparameters for MuJoCo environments, TD3 used the same (common) hyperparameters.

In SPP-SAC implementation we used the following architectures:

- Actor: $\dim(\mathcal{S}) \longrightarrow 256$ ReLU $\longrightarrow 256$ ReLU $\longrightarrow \dim(\mathcal{A})$ tanh;

- Critic: $\dim(\mathcal{S}) + \dim(\mathcal{A}) \longrightarrow 256$ ReLU $\longrightarrow 256$ ReLU $\rightarrow 1$;

- CM: $2 \cdot \dim(\mathcal{S}) \longrightarrow 64$ tanh $\longrightarrow 32$ tanh $\longrightarrow \dim(\mathcal{A})$ tanh;

| Hyperparameter | Value |
|---|---|
| $\gamma$ | 0.99 |
| $\tau$ | 0.005 |
| actor/critic learning rate $l_\phi = l_\theta$ | 0.0002 |
| $\lambda$ learning rate $l_\lambda$ | 0.0001 |
| episode length | 5000 |
| batch size | 100 |
| test episodes | 10 |
| update freq. & grad.steps. | 50 |
| exploration noise param. | 0.1 |
| policy noise | 0.2 |
| noise clip. | 0.5 |
| replay buffer size | same as number of env. interacts |
| parameters specific for SPP-TD3 | |
| $d$ state consistency | 0.2 |
| CM hyper-parameters | |
| init. rand. samples | 400000 |
| learning rate | 0.001 |
| batch size | 128 |
| update frequency | 500 steps |
| update epochs | 1 |
| update batches | 200 |

Table 4: SPP-TD3 Algorithm hyperparameters for SafetyGym environments. Observe that for this environment specifically we used larger number of CM pretrain samples

| Hyperparameter | Value |
|---|---|
| $\gamma$ | 0.99 |
| $\tau$ | 0.005 |
| actor/critic learning rate $l_\phi = l_\theta$ | 0.0001 |
| $\lambda$ learning rate $l_\lambda$ | 0.0001 |
| episode length | 5000 |
| batch size | 100 |
| test episodes | 10 |
| update freq. & grad.steps. | 50 |
| exploration noise param. | 1 |
| policy noise | 0.2 |
| noise clip. | 0.5 |
| replay buffer size | 250000 |
| parameters specific for SPP-TD3 | |
| $d$ state consistency | 0.2 |
| CM hyper-parameters | |
| init. rand. samples | 100000 |
| learning rate | 0.001 |
| batch size | 128 |
| update frequency | 500 steps |
| update epochs | 1 |
| update batches | 200 |

Table 5: SPP-TD3 Algorithm hyperparameters for AntPush environment. Observe that for this environment specifically we used fixed buffer size which is significantly smaller than the total number of env interacts, and a much larger exploration noise param equal to $1$.

| Hyperparameter | Value |
|---|---|
| $\gamma$ | 0.99 |
| $\tau$ | 0.005 |
| actor/critic learning rate $l_\phi = l_\theta$ | 0.001 |
| $\lambda$ learning rate $l_\lambda$ | 0.0001 |
| batch size | 100 |
| test episodes | 10 each 1000 frames |
| update freq. & grad.steps. | 50 |
| $\alpha$ | 0.2 |
| $\alpha$ learning rate | 0.001 |
| replay buffer size | $1e6$ |
| parameters specific for SPP-SAC | |
| $d$ state consistency | 0.2 |
| CM hyper-parameters | |
| init. rand. samples | 10000 |
| learning rate | 0.001 |
| batch size | 100 |
| update frequency | 1000 steps |
| update batches | 100 |

Table 6: SPP-SAC Algorithm hyperparameters

# E  ADDITIONAL EXPERIMENTS

## E.1  ABLATION STUDY

We performed a thorough ablation study of the features that we implemented in the SPP-TD3 algorithm presented in Sec. C. Ablation study was performed in Ant (Fig. 7a) and Doggo-Goal (Fig. 7b). We explain the meaning of particular ablations visible in Fig. 7.

- *SPP-TD3 less init. sampl.* means that less randomly generated experience the algorithm added to the replay buffer at the beginning of executing Alg. 3 (the first *repeat until* block). In some cases, like SPP-TD3 for Doggo-Goal, see Table 4 we choose to include a large number of random samples in the buffer (400000). However, this has no considerable effect on performance. In fact, the SPP-RL agent utilizing the number of random samples the same as vanilla RL performs better initially. Only in the long term, slightly losing to the agent utilizing more experience. This shows that the performance boost of SPP-RL still persists when fed the same number of initial random samples as in the corresponding vanilla RL method.

- *SPP-TD3 w/o state consistency* the state consistency defined in Property 1 is not introduced in the algorithm; therefore, there is no incentive on the policy to choose as targets realizable states. Therefore it demonstrates that the SPP approach is more than just a policy reparametrization.

- *SPP-TD3 $Q(s, s')$ critic (state-state critic)* replaced the traditional target $Q$ function computation using state-action pairs with state-next state pairs. Hence the step of computing target actions is omitted. As a result, the algorithm did not converge in several cases and exhibited a significant variation of the obtained returns between different runs. This is consistent with the observed performance of D3G algorithm in Ant (Fig. 3b). We do not show this ablation in Doggo-Goal, as the algorithm did not converge in this environment.

- *SPP-TD3 w/o state normalization* $\pi$ is not being normalized using the current mean and min/max values of the past observations in replay buffer $\mathcal{D}$.

- *SPP-TD3 const. $\lambda = 1$* the Lagrangian multipliers method is not used. The algorithm uses the constant value $\lambda = 1$ when performing policy updates. We chose this particular $\lambda$ value, as it tends to work well in Ant, whereas in Doggo-Goal results in lack of convergence; this shows how delicate the matter of choosing appropriate $\lambda$ (when fixed) per given environment is.

- *SPP-TD3 const.* $\lambda = 0.1$ *or* $\lambda = 0.5$ the Lagrangian multipliers method is not used, the algorithm uses the constant value $\lambda = 0.1$ or $= 0.5$ respectively when performing policy updates. The particular $\lambda$ values resulted in much better performance in Doggo-Goal than $\lambda = 1$ (which works in Ant). Observe that $\lambda = 0.1$ results in even better performance than the base SPP-TD3 implementation utilizing Lagrangian multipliers. However, employing the Lagrange multipliers provides a natural way of solving the constrained objective optimization, and the hyper-parameter $d$ is adjusted once for all the environments.

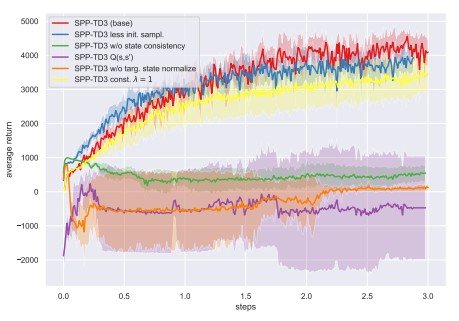

(a) Ablation study using Ant

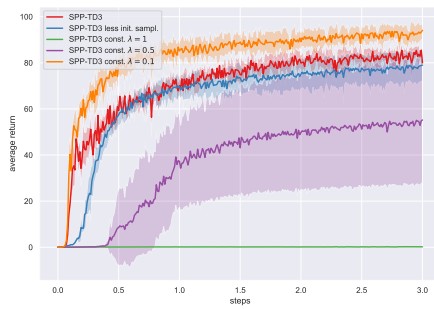

(b) Ablation study using Doggo-Goal

Figure 7: Ablation study of various SPP-TD3 algorithm features, performed using Ant and Doggo-Goal environments. For the exact features of the algorithm that were tested refer to the discussion in Appendix E.1. Figures show test return computed every 5k frames averaged over 5 different seeds. The continuous curve is the mean, whereas the faded color regions std. deviation.

## E.2 SPP-RL vs Vanilla RL Replay Buffer

As argued in Sec. 5.4 the performance boost visible in SPP-RL vs. vanilla RL approaches is presumably due to more efficient exploration performed by SPP-RL algorithms vs. vanilla RL approaches. One empirical argument given concerns the performance of the so-called TD3 shadow agent, i.e., vanilla TD3 agent implementation utilizing (partially) the experience gathered by SPP-TD3 agent to perform its Actor&Critic updates. Here we provide empirical evidence that the distribution of observations in replay buffers gathered by SPP-RL and vanilla RL implementations differs considerably. In Fig. 8 we present an empirical study of distributions of states gathered in SPP-SAC and vanilla SAC replay buffers for the Doggo-Goal task. The state-space in this task has 72 dimensions. Hence to make it amenable to visual investigation and entropy computation, we encode the state vectors using a random encoder. The encoder architecture that we used for this task is just initialized (default PyTorch ini) and not trained architecture: $72 \rightarrow 20 \tanh \rightarrow 10 \tanh \rightarrow 2$. Fig. 8 show plots of discretized state distributions gathered by example run of vanilla SAC and SPP-SAC respectively using the Doggo-Goal environment and encoded using the random encoder. We also compute cross-entropy to quantify the difference between those distributions as the training of both algorithms progresses and the replay buffer is being filled up. Observe that the cross-entropy is clearly increasing as the replay buffer is being filled up, which suggests that vanilla RL and SPP-RL algorithms gather different observation distributions in the replay buffer.

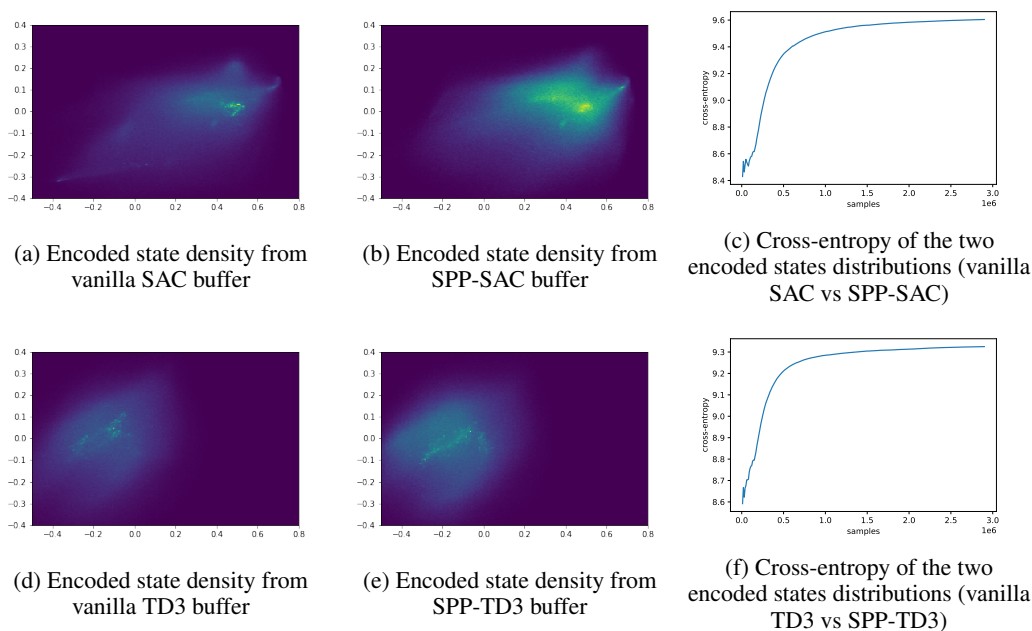

(a) Encoded state density from vanilla SAC buffer

(b) Encoded state density from SPP-SAC buffer

(c) Cross-entropy of the two encoded states distributions (vanilla SAC vs SPP-SAC)

(d) Encoded state density from vanilla TD3 buffer

(e) Encoded state density from SPP-TD3 buffer

(f) Cross-entropy of the two encoded states distributions (vanilla TD3 vs SPP-TD3)

Figure 8: Visualizations of states distribution (density histogram plot) in the replay buffer at the end of training taken from single vanilla SAC, TD3, SPP-SAC, and SPP-TD3 runs. States are encoded in 2D using the random encoder. Figs. 8c,8f is the cross-entropy of the two discrete state distributions (vanilla vs SPP) with respect to the quantity of experience in the replay buffer (excluding the initial random initialization).

