# OpenReview forum: "SPP-RL: State Planning Policy Reinforcement Learning"
_ICLR.cc/2022/Conference — ICLR 2022 Submitted_

### Official Review · Reviewer_oVKQ · 2021-10-28

**Correctness:** 3
**Technical Novelty And Significance:** 3
**Empirical Novelty And Significance:** 3
**Recommendation:** 8
**Confidence:** 4

**Main Review:**

The authors mention that, during learning, if the buffer of the
samples is full new samples are not added anymore. Why is that? It is
just for solving the ICM or also for learning Q and \pi? Please
clarify.

Apparently there were some implementation issues with D3G, otherwise it
is not clear why there was a big difference in its performance, given
that D3G follows a similar approach. Why the authors did not try to
re-implement D3G in their environment?

It is not clear why SAC-vanilla behaves better than SPP-SAC in the
car-push domain?

If would be interesting to discuss how to include other constraints,
e.g., a cost function for violating safety, within the proposed
approach.

The authors argue, and experimentally show, that a state-state mapping
can perform better exploration than a state-action mapping, but it is
not clear why this is the case. Why there is a better exploration?

The parameter "d", which regulates the state consistency, is quite
relevant to the proposed approach, however, the authors did not show
how different values of "d" affect the results.

It seems that the transition function is assumed to be
deterministic. How much this affects the applicability of the proposed
approach?

The paper in the introduction talks about data efficiency and
interpretability, however, it is not clear how is interpretability
improved by the proposed approach?

Typos:
- build on The => build on the
- caption Fig 5 c): ... bule ... => ... blue ...
- however, we if we show


**Summary Of The Paper:**

The paper describes an approach where a state-state mapping is
learned (called state-planning policy or SPP) coupled with an
off-policy RL system (the authors included experiments with DDPG, TD3,
and SAC).

Learning a state-state mapping, Q(s,s'), needs also to learn
an inverse dynamic model to estimate which action (a) will take the
current state (s) to the next state (s').

The authors framed the problem as a constrained optimization approach,
where the objective is to maximize the sum of discounted rewards such
that the differences between the target states (produced by the
policy) and the actual states (produced by the environment with the
action from the control model) are within a certain threshold
distance. In particular, the authors solved the optimization problem
via the Lagrange multiplier method.

Experiments were performed in different domains and with different
state-of-the-art DRL systems.


**Summary Of The Review:**

The paper describes a new approach for learning a state-state mapping.
Pros:
- It is a well presented and clearer approach for learning a state planning
policy, than previous work (in particular, D3G)
- It opens a new research avenue in terms of exploration comparisons between
state-state and state-action policies
Cons:
- An alternative formulation of a previously published idea (D3G) which can be
seen as an incremental contribution
- Not completely clear why and under which circumstances the state-state
mapping is preferred over the state-action mapping
- Lack of assessment on the performance of the approach with different
state consistency threshold values (b)

---

> ### Author Response · Authors · 2021-11-17
> **Thank you for appreciating our results and remarks, we answer your questions**
>
> > _The authors mention that, during learning, if the buffer of the samples is full new samples are not added anymore. Why is that? ..._
>
> A: We use the same replay buffer size for all of the trainings performed within given environment. We fix the replay buffer size and stop adding new experience when filled, to make the comparison plots fair in cases where we put a lot of random initial samples into the buffer (like in SPP-TD3 Doggo environments). We do not take advantage of pretrained IDM models before initiating Actor&Critic training loop.
> BTW observer that SPP algorithms  perform well even in case the replay buffer contains mainly off-policy samples.
>
> > _Apparently there were some implementation issues with D3G, otherwise it is not clear why there was a big difference in its performance..._
>
> A: We did not re-implement D3G in full. Authors of D3G shared their code tested in MuJoCo environments. Observe that the big difference is already coming from employing Q(s,a) (like in our approach) instead of Q(s,s')  (like in D3G), this is shown in the ablation study. Also D3G utilizes one additional model (forward dynamics model) and the cycle loss, which increase complexity of the overall algorithm as compared to ours.
>
> > _It is not clear why SAC-vanilla behaves better than SPP-SAC in the car-push domain?_
>
> A: We apologize for the typo, red and green colors were replaced only in Car-Push plot. As expected SPP-SAC beats SAC-vanilla also in this domain. It is now corrected in the revised version of the paper submitted here.
>
> > _If would be interesting to discuss how to include other constraints, e.g., a cost function for violating safety, within the proposed approach._
>
> A: One way, on which we work on, is to implement a safety critic for prediction of the expected cost of the policy trajectory. Then enforce the constraints on the policy by introducing another dual variable (Lagrange multiplier) and solve the policy optimization using the new Lagrangian. We will add a comment on that in the revised paper.
>
> > _The authors argue, and experimentally show, that a state-state mapping can perform better exploration than a state-action mapping, but it is not clear why this is the case. Why there is a better exploration?_
>
> A: We are working on a precise theoretical support of the claims, but for now offer only intuition. Exploration using SPP policies results in more viable experience being collected in the replay buffer, leading to more efficient Actor & Critic training.
> The constrained optimization and state-planning policies help the agent find high return trajectories by performing exploration in the target-state space. These claims are experimentally supported by showing the improved performance of a TD3 shadow agent (over vanilla TD3) when utilized experience from SPP-TD3 agent replay buffer (Sec. 5.4.), and the replay buffers cross entropy computation (Sec. E.2.).
>
> > _The parameter "d", which regulates the state consistency, is quite relevant to the proposed approach, however, the authors did not show how different values of "d" affect the results._
>
> A: Observe that, in our case, the constraint has a clear physical interpretation of the policy target-states' consistency (how much they can differ from the physical next-states). As we normalize, the observations for the constraint computation are set once-for-all for the studied environments. We find that worked well in all studied environments. Of course, there is a trade-off here. We could trade target-state consistency for slightly better returns in some of the envs (e.g. Doggo-Goal shown in the ablation study in Section E.1), but we believe that keeping the policy prediction physical is an essential feature of our algorithm.
>
> > _It seems that the transition function is assumed to be deterministic. How much this affects the applicability of the proposed approach?_
>
> A: At present we limit our scope of applications to robotic continuous environments based on a rigid-body dynamics model. Under this assumption we were able to derive the convergence of Q-learning embedded in our algorithm. However, our algorithm could be applied in principle to other environments including discrete grid-world like and environments with probabilistic transition function.
>
> > _The paper in the introduction talks about data efficiency and interpretability, however, it is not clear how is interpretability improved by the proposed approach?_
>
> A: Example regarding the reliability: SPP approach finds the good solution more reliably with respect to random seed as compared to HIRO Fig. 5 a), and interpretability aspect illustrated in Fig. 5 b), c) & d) where the policy planned path in the state-space for the bad and good solution can be visually distinguished. Moreover, training of state-planning policies is desirable as they operate on a higher level than traditional policies (plan a trajectory in the state-space) and have been employed in other RL algorithms like HIRO and D3G.

---

> > ### Comment · Reviewer_oVKQ · 2021-11-30
> > **Comments after rebuttal**
> >
> > I have read all the reviews and the authors response, and I still think this paper is worth publishing. In particular, it opens an interesting discussion about the possible advantages between state-state and state-action policies (although they are not properly addressed in the paper).

---

### Official Review · Reviewer_B8gx · 2021-11-02

**Correctness:** 4
**Technical Novelty And Significance:** 2
**Empirical Novelty And Significance:** 2
**Recommendation:** 3
**Confidence:** 4

**Main Review:**

Since this is simply a "new algorithm" paper, it is evaluated along three axes: 1) Empirical results, 2) Theoretical results 3) Insights into why it works and when to use it. For 1) the gains for the doggo environment seem strong. However, that is about it. I do not think it is sufficient to propose a new RL algorithm and solely have results for one environment from a proprioceptive state as evidence that it works. There are no pixel based environments, which the community is currently focused on, and the MuJoCo experiments are weak for both TD3 and SAC, showing little gains despite increased wall clock time. 2) There is no theory, which is fine, but adds emphasis to 1). Finally for 3) I am not convinced at all this method makes sense based on the analysis provided. I don't read this and feel like I really understand why it should work and where I should expect it to work. Given this, my score is a reject. I would be willing to increase my score if there are improvements along these three axes, most likely 1) and 3). For example, an additional set of experiments + intuition about why it works/doesn't work in those particular experiments.

More detailed comments:
* The opening paragraph makes claims about poor sample efficiency and interpretability of RL algorithms, but does not cite any papers for this. Then the paper itself does not really address this.
* The performance does not seem particularly impressive in Figure 3 - SPP basically adds nothing to TD3 and SAC, aside from one bad seed in TD3 for Humanoid. However, the actual numbers reported are lower than usually seen for Humanoid. It is common to see ~5k after 1M steps and ~6k after 2M steps for both TD3 and SAC. Given this, the results seem quite weak, and given that this is an empirical paper, it is a significant negative.
* There are no pixel based tasks. A few years ago this criticism would have been unfair, but nowadays the community is very focused on this setting, and there are many RL algorithms that could be built on top of with this type of approach, for example RAD uses a similar off policy algorithm.

* After reading the paper I don't really get *why* I would use this method. It takes a longer time to run, and doesn't outperform in many cases. What is it about the doggo environment that makes it work better? The comment "due to improved exploration in some sense" shows that even the authors do not know why their method did well on the environments where it did. I see the plots in Section E.3 and it does seem to collect more different states, but this would be expected with a stronger policy anyway, I still don't know why this is the case. For example, the ablation study has very different ordering for the different components for Ant and Doggo, why? It isn't clear to me at all.
* How were the hyperparameters chosen? I see them listed, but there is no discussion of how they were tuned.
* It is great to see the method run for ten seeds, and with transparency about the wall clock time. I will say though that the wall clock time is definitely a negative of the method which may limit applicability. How does the wall clock time scale for higher dimensional problems? What is the bottleneck that causes this doubling in time for the environments considered?

Minor comments/typos:

* In the first line of the paper it says RL is a "machine learning technique". This may be picky, but I am not sure it is a great way to describe RL. RL is both a problem (maximizing reward in an MDP) and a class of methods.
* In the "summary of results" section change "overperform" → "outperform".
* "Related Works" → "Related Work".
* Bottom of p3, "Obviously, SPP approach to work requires" → "in order to work, SPP requires".
* Sec 4, "details of SPP algorithm" → "details of *the* SPP algorithm". "*the* Spinning Up RL online resource". "*the* DDPG algorithm".
* Sec 5, "we performed a set of benchmarks" → "we performed experiments on a set of benchmarks". "overall sample efficiency of RL procedure" → "overall sample efficiency of *the* RL procedure".
* Sec 5.4, first sentence doesn't make sense.
* The use of apostrophes/speech marks for "shadow agent" is incorrect.

**Summary Of The Paper:**

This paper proposes a new RL algorithm whereby the policy selects a new state rather than action, with constraints to ensure the next state selected is a valid one. The contribution is the new algorithm "SPP-RL" which can be applied to off policy algorithms such as TD3/SAC. There is experimental evidence this may be an effective approach in some settings.

**Summary Of The Review:**

This is an empirical RL paper introducing a new algorithm, but the results are not convincing either based on their strength or intuition. At present the paper does not convince the reader to use this method or build upon it, thus it does not meet the bar for acceptance.

---

> ### Author Response · Authors · 2021-11-16
> **Thank you, we answer all of your concerns, part 1**
>
> First of all, thank you for the review. You raised a few important remarks. Let us respond to the main reviewers' critique summarized in the first paragraph. Our main motivation has been to explore and propose new algorithms for reliable and interpretable reinforcement learning. We think the sample complexity (compared to model-based approaches) and policy interpretability of model-free approaches are far from satisfactory even in simple continuous state-vector-based environments.
>
>
> But, we find your comments
>  _the gains for the doggo environment seem strong. However, that is about it,_
> and
> _I do not think it is sufficient to propose a new RL algorithm and solely have results for one environment_
>
> being unfounded. Can you make more precise 'solely have results for one environment'? There are papers studying just the Acrobot system... Here we have three varied Doggo environments (goal, button & columns), and what about other environments where SPP algorithms outperform: Ant & Humanoid in case of DDPG, CarPush (TD3 & SAC), and AntPush (HIRO). You also ignore the reliability: SPP approach finds the good solution more reliably with respect to random seed Fig. 5 a), and interpretability aspect illustrated in Fig. 5 b), c) & d) where the policy planned path in the state-space for the bad and good solution can be visually distinguished.
>
> > Q: _Finally, for 3) I am not convinced at all this method makes sense based on the analysis provided. I don't read this and feel like I really understand why it should work and where I should expect it to work._
>
> > Q: _After reading the paper I don't really get why I would use this method. It takes a longer time to run, and doesn't outperform in many cases. What is it about the doggo environment that makes it work better?_
>
> A: We proposed a new RL algorithm employing state-planning policies. It is a simpler approach than other related RL methods employing state-planning policies (HIRO and D3G). There is a significant performance boost of off-policy RL learning in the tested Safety-Gym robotic locomotion tasks. SPP clearly outperforms its vanilla RL counterpart in three varied Doggo environments (Goal, Button & Columns), Ant & Humanoid for vanilla DDPG, CarPush (TD3 & SAC), and AntPush (HIRO). Reliability: SPP approach finds the good solution more reliably with respect to random seed Fig. 5 a), and Interpretability: Fig. 5 b), c) & d) the policy planned path in the state-space for the bad and good solution can be visually distinguished.
>
> Moreover, training of state-planning policies is desirable as they operate on a higher level than traditional policies (plan a trajectory in the state-space) and have been employed in other RL algorithms like HIRO and D3G.
>
> What makes it work better? Exploration using SPP policies results in more viable experience being collected in the replay buffer, leading to more efficient Actor & Critic training. It is also possible that the constrained optimization and state-planning policies help the agent find high return trajectories by performing exploration in the target-state space. These claims are experimentally supported by showing the improved performance of a TD3 shadow agent (over vanilla TD3) when utilized experience from SPP-TD3 agent replay buffer (Sec. 5.4.), and the replay buffers cross entropy computation (Sec. E.2.).
>
> > Q: _2) There is no theory, which is fine, but adds emphasis to 1). Finally for 3)_
>
> A: The theoretical results are contained in Section B and include the policy gradient theorem for the studied Lagrangian objective and convergence of Q-learning in the SPP setting (TD3 like). The further theoretical derivations like the convergence for actor-critic we will pursue as future work.

---

> ### Author Response · Authors · 2021-11-16
> **Thank you, we answer all of your concerns, part 2**
>
> >  _The performance does not seem particularly impressive in Figure 3 - SPP basically adds nothing to TD3 and SAC, aside from one bad seed in TD3 for Humanoid..._
>
> A: Observe in the same Figure, the performance of SPP is significantly better in the case of DDPG in Humanoid. Also, SPP matching vanilla off-policy in environments with 1d locomotion tasks (just learn to walk forward) is still surprising. The search for SPP policy (mapping states to target-states) is performed in a much larger space than in the case of vanilla policy (mapping states to actions). For comparison in Humanoid, the dimension of the state space is $376$, whereas the dimension of the action space is $17$.
>
> >  _I see the plots in Section E.3 and it does seem to collect more different states, but this would be expected with a stronger policy anyway_
>
> A: Yes, perhaps, but a stronger policy couldn't be trained using vanilla off-policy in this case…
>
> >  _There are no pixel based tasks. A few years ago this criticism would have been unfair, but nowadays the community is very focused on this setting..._
>
> A: We haven't got results for pixel-based tasks yet, but we also don't like to abandon state-vector based envs yet. Actually, Our current work-in-progress is pixel-based environments, and we are convinced that the presented approach can be successfully applied there. The approach is based on using the idea of unsupervised image encodings learned by the Inverse Dynamic Model like in the curiosity paper D. Pathak et. al. 2017. Reparametrize the policy to predict the target-state embeddings just like the physical states in the current setting.
>
>
> > _The opening paragraph makes claims about poor sample efficiency and interpretability of RL algorithms, but does not cite any papers for this. Then the paper itself does not really address this._
>
> A: We will make this more precise with additional citations in the revised version. However, these are well-known observations in the ML community. Model-free approaches are significantly less sample efficient than model-based approaches in continuous environments. See for example [1]. Interpretability and reliability aspects are due to thinking of AI in general as a black-box, see [2], [3].
> [1]Michael Janner and Justin Fu and Marvin Zhang and Sergey Levine. When to Trust Your Model: Model-Based Policy Optimization. NeurIPS '19.
> [2] Marcus, G., and Davis, E. (2019). Rebooting AI: Building Artificial Intelligence We Can Trust. Pantheon.
> [3] P. Henderson, R. Islam, P. Bachman, J. Pineau, D. Precup, D. Meger. Deep Reinforcement Learning that Matters. AAAI-18
>
> > Q: _For example, the ablation study has very different ordering for the different components for Ant and Doggo, why?_
>
> A: We tested different algorithm features in the provided ablation study in the two envs. Both of the studies have the _SPP-TD3 less init. sampl._ to show that SPP-TD3 works when the number of initial random samples is the same as in vanilla TD3, and _SPP-TD3 const. $\lambda = 1$_ is included to show that SPP-TD3 with fixed $\lambda$ can perform well in Ant, whereas in Doggo-Goal does not converge. The ablation study in Ant tests different features of the algorithm to show all of them are crucial for the performance, whereas the study in Doggo-Goal shows that the performance of SPP-TD3 with fixed $\lambda$ varies greatly, so adjusting $\lambda$ per env can be a delicate issue, which motivates the implemented Lagrangian optimization approach.
>
> > Q: _How were the hyperparameters chosen? I see them listed, but there is no discussion of how they were tuned._
>
> A: We just performed a hyper-parameter optimization using a simple grid search. After a series of initial experiments, we predefined a list of possible values per each parameter. Using SPP-TD3 for example, we tried learning rates $\{0.001, 0.0005, 0.00001}$, expl. noise $\{0.1, 0.2\}$, init rand. samples $\{25k, 50k, 100k, 200k, 400k\}$, CM upd. freq. $\{500, 1k, 2k\}$, CM upd. batches $\{100,200\}$. The performance can be clearly improved further using more advanced optimization techniques. We will include the exact considered values in the paper revision.
>
> > Q: _How does the wall clock time scale for higher dimensional problems? What is the bottleneck that causes this doubling in time for the environments considered?_
>
> A: Ant is a fairly high-dimensional problem ($111$ dim. state-vectors). The bottleneck is the lack of parallelism of our implementation. We were in the first place interested in evaluating the approach rather than optimizing the code. Clear improvements can be achieved by a parallel training of the actor, critic, and IDM. Also, in case of even higher-dimensional problems (pixels) we rather make our algorithm work within a space of state embeddings.

---

> > ### Comment · Reviewer_B8gx · 2021-11-22
> > **Still not convinced**
> >
> > Thank you for responding to my comments.
> >
> > I think some of these clarifications will improve the paper, so it would be great if they could all be included - especially the choice of hyperparameters.
> >
> > Fundamentally, the main issue I have is still there, it only seems to actually improve DDPG, which has not been state-of-the-art since 2017. There are basically no gains when applied to SAC or TD3, which are the most commonly used methods for continuous control from proprioceptive states. Finally, the paper is purely empirical but there are no pixel based tasks, which are primarily used these days for empirical papers at venues such as ICLR. There is no evidence that this method would scale for example to latent states.
> >
> > I do not think it is possible to address these issues in the limited time for the rebuttal, so will likely remain with my current score.

---

> > > ### Author Response · Authors · 2021-11-23
> > > **Dear Reviewer B8gx**
> > >
> > > Let us take this opportunity to once again reiterate our points.
> > >
> > > * We are willing to include the suggested improvements in the paper next time we'll have an opportunity to do so;
> > >
> > > * DDPG is a solid base of the TD3 and SAC algorithms, not being any more state-of-the-art. DDPG can be understood as a continuous Q-learning, and TD3 and SAC build on that, so we think it is still valid to claim an improvement in the vanilla DDPG case;
> > >
> > > * Improving existing state-of-the-art on Humanoid and Ant from proprioceptive states was not our main objective, as our approach is clearly advantageous (as shown) for problems involving spatial robotic locomotion on a 2D arena, safety-gym suite and AntPush in our case, not only learning to run forward 'flat' as fast as possible;
> > >
> > > * we are happy that the reviewer finds pixel-based tasks the most suitable for ICLR, but we hope for that the community to be inclusive, and find appealing and suitable also more fundamental reinforcement learning problems in continuous control from proprioceptive states not so hungry for computational resources... The main reason we find continuous control from proprioceptive states appealing is the still persisting sample efficiency, reliability, and interpretability issues. Some problems involving robots with multiple joints and complicated planning are just hard to realize in the pixel-based setting.

---

### Official Review · Reviewer_F9WZ · 2021-11-02

**Correctness:** 2
**Technical Novelty And Significance:** 2
**Empirical Novelty And Significance:** 2
**Recommendation:** 5
**Confidence:** 3

**Main Review:**

While the problem is of interest, I have several concerns about this work.

- The treatment of the constrain in the problem is lacking formality, which makes it difficult to evaluate some of the authors contributions.

a) The dual variable \lambda needs to be projected to the non-negative orthant in Algorithm 1, 2 and 3.

b) The author's algorithm takes the form of a primal-dual algorithm for CMDPs. In this regard and due to the non-convexity of the Lagrangian, the choice of both the primal and dual step sizes plays a critical role in guaranteeing convergence. The authors completely ignore this issue (there is no dual step size in their algorithm). See e.g., [A].

c) Some of the analysis in Appendix B does not make much sense. For example, section B.2 presents a policy gradient theorem, but this is not the gradient that the authors take in Algorithm 1 (the gradient is taken with respect to the Lagrangian). In general the analysis disregards the effect of the dual variables.

- The hyperparameter d in the optimization problem (1) might be difficult to design. How sensitive are the results to its value? How to handle infeasibility in the problem?

Minor comments:
	- Several references are going out of margins.

[A] Borkar, Vivek S. "An actor-critic algorithm for constrained Markov decision processes." Systems & control letters 54, no. 3 (2005): 207-213.


**Summary Of The Paper:**

This work presents a a new reinforcement learning algorithm in which planning is performed purely on the state space instead of the more common state-action space. A constrained optimization problem is formulated which is solved exploiting duality.

**Summary Of The Review:**

In its current form, I cannot recommend this paper for publication.

---

> ### Author Response · Authors · 2021-11-16
> **Thank you for important remarks, we answer all of your concerns**
>
> First of all, thank you for the important remarks that helped to improve the paper. You are right that our policy optimization is a primal-dual algorithm. This has been clarified in the new paragraph we added in the revised version (submitted here) of the paper on p. 17. We emphasize that the paper's goal is not to present a new algorithm for CMDP, but rather a new algorithm for deep reinforcement learning in continuous environments where the policy is reparametrized to plan the target-states instead of actions. We find the constrained optimization formulation as a convenient way of guaranteeing the feasibility of the policy target-states. We experimentally show that the resulting SPP-RL algorithm is characterized by improved exploration and outperforms vanilla off-policy algorithms in a few continuous environments (safety-gym suite and AntPush). Also is significantly simpler than existing related algorithms (HIRO, D3G).
>
> >  _The treatment of the constrain in the problem is lacking formality, which makes it difficult to evaluate some of the authors contributions_
>
> A: We added a remark about the relation of our approach to the primal-dual algorithm in the revised version of the paper. Our goal has been to first propose a new algorithm for deep off-policy continuous RL and validate it experimentally. The provided theoretical Section is a preliminary work that will be extended, including proofs of actor&critic RL algorithm convergence, which we find as an exciting avenue of future research on SPP-RL.
>
> > a) _The dual variable \lambda needs to be projected to the non-negative orthant in Algorithm 1, 2, and 3._
>
> A: We explained this in the paper (beginning of p. 6) _To ensure that we perform minimization equation 6
> within the domain of positive λ values, we optimize the parameter of the softplus function_
>
> > b) _The author's algorithm takes the form of a primal-dual algorithm for CMDPs. In this regard and due to the non-convexity of the Lagrangian, the choice of both the primal and dual step sizes plays a critical role in guaranteeing convergence_
>
> A: Yes, thank you for this remark. In fact, there is a typo in the Algorithm presentation; we forgot to mention the dual variable learning rate in the algorithm explicitly. It was used in the code (which can be checked), and as the reviewer observed, the learning rate needs to be appropriately adjusted for convergence. This is now fixed in the revised paper (submitted here). We included a note. There are three learning rates in total: actor, critic, and dual variable ($l_\lambda$). We adjusted them by a hyper-parameter optimization. The actor and critic l.r. were the same; the values are given in the hyper-parameter table and $l_\lambda=0.0001$, which is not larger, and usually smaller than l.r. of actor&critic, exactly like the theory predicts!
>
> > c) _Some of the analysis in Appendix B does not make much sense. For example, Section B.2 presents a policy gradient theorem, but this is not the gradient that the authors take in Algorithm 1 (the gradient is taken with respect to the Lagrangian). In general, the analysis disregards the effect of the dual variables._
>
> A: We want to clarify this important concern. In Section B.2 where we compute the policy gradient, we omitted computation of the full gradient, including the dual part. We apologize for this omission. We include a new paragraph with the remaining straightforward computation of the constraint function gradient in the revised paper (submitted here) on p. 15. The gradient w.r.t. to the dual variable is obvious and hence omitted.
>
> Regarding Section B.3. concerning the convergence of Q-learning. We establish the convergence of Q-learning in the classical setting, under the assumption that (explorative) SPP policy outputs realizable target-states and the environment is based on a deterministic rigid-body dynamics model. Our implementation is consistent, and we use the classical Q-learning scheme like in TD3 and SAC.
>
> > Q: _The hyperparameter d in the optimization problem (1) might be difficult to design. How sensitive are the results to its value? How to handle infeasibility in the problem?_
>
> A: We do not agree that $d$ is difficult to design. Observe that, in our case, the constraint has a clear physical interpretation of the policy target-states' consistency (how much they can differ from the physical next-states). As we normalize, the observations for the constraint computation $d$ are set once-for-all for the studied environments. We find that $d=0.2$ worked well in all studied environments. Of course, there is a trade-off here. We could trade target-state consistency for slightly better returns in some of the envs (e.g. Doggo-Goal shown in the ablation study in Section E.1), but we believe that keeping the policy prediction physical is an essential feature of our algorithm.

---

### Decision · Program_Chairs · 2022-01-20

**Decision:**

Reject

**Comment:**

The paper presents a RL planning algorithms where a policy selects a reachable state. The empirical evaluation shows promising results in some environments. While all the reviewers agreed that the state planning RL is a relevant and promising direction, the reviewers expressed concerns with the rigor, significance of the results, and incremental novelty.

To improve them paper the authors should:

- Bring the theoretical foundation in the main text, and add more rigorous analysis, including the limitations of the method.
- The readability of the figures needs to be improved. The legend on the figures is too small and colors are too similar that renders the figures unreadable and confusing.
- If the authors' goal is to develop a method for interpretable RL, then some results and analysis need to address the interpretability of method.